# Understanding the Parameter Space Geometry of Transformers Encoding Boolean Functions

Blanka Kövér [1]    Alexandra Butoi [2]    Anej Svete [2]    Michael Hahn [3]    Ryan Cotterell [2]

## Abstract

Transformers consistently fail to learn certain simple functions that are provably expressible with specific parameter settings. This gap between *learnability* and *expressivity* is particularly prominent for sensitive functions—functions whose output is likely to change if a single bit of the input is flipped—for example, PARITY. While prior work has established that transformers exhibit a bias toward functions with low average sensitivity, the precise mechanism underlying this bias remains poorly understood. To shed light on this phenomenon, we study the geometry of transformers' parameter space. We show that sensitive functions—even when representable—occupy a vanishingly small region that random initialization is very likely to miss. Specifically, we shift the focus from average sensitivity to the full sensitivity profile—the distribution of sensitivity values across all inputs—and prove that randomly initialized transformers almost surely compute functions which have low-sensitivity strings. Consequently, any function that lacks such strings is provably unlearnable.

## 1. Introduction

Much theoretical work on transformers aims to better understand their abilities and limitations. A central tool for this is studying their *expressivity*, i.e., for which functions there exist parameter settings that allow a transformer to compute them. See Strobl et al. (2024) for a recent survey.

Expressivity alone, however, is not enough to predict what transformers can learn in practice: even when the right parameter settings exist, they may not be reachable through training, which is precisely the question of *learnability*.

A canonical example of the difference between expressivity and learnability is PARITY, the function which returns whether the input has an even or odd number of ones. With layer norm, transformers are expressive enough to represent it (Chiang & Cholak, 2022), yet fail to learn it in practice (Bhattamishra et al., 2023; Butoi et al., 2025). A generalization of this finding has emerged from related empirical and theoretical work (Bhattamishra et al., 2023; Abbe et al., 2023; Hahn & Rofin, 2024): transformers struggle to learn Boolean functions that are *sensitive*, i.e., those for which flipping a single input bit is likely to change the output. However, a precise connection between sensitivity and what optimization can find is still missing; we take an initial step in this direction by analyzing the *volume* of parameter space that different classes of Boolean functions occupy.

Throughout, we assume a soft-attention transformer, which we treat as a *recognizer* of a formal language—a function that classifies input strings as belonging to the language or not—over a binary alphabet. Our analysis is divided into two cases, corresponding to the two settings of the layer-norm stabilizing constant $\epsilon$ in the denominator of Eq. (13), which prevents division by zero. The first is a warm-up: when $\epsilon$ is bounded away from zero, the layer-norm denominator stays bounded below and the whole transformer is uniformly Lipschitz in its parameters. Even more sharply, we show that no non-constant Boolean function is recognizable for arbitrarily long inputs (Cor. 3.4). This suggests that, when $\epsilon > 0$, transformers are deeply inexpressive as models of computation.

Our main contribution is our analysis of the second case, $\epsilon = 0$, where the layer-norm denominator can become arbitrarily small and the per-layer Lipschitz constant is no longer uniformly bounded. We provide a general measure-theoretic framework that generalizes and strengthens Hahn & Rofin's (2024) results on sensitive functions. In particular, we shift the focus from average sensitivity to a function's *sensitivity profile*, i.e., the distribution over input strings of each possible sensitivity value. The sensitivity profile captures more

[1]Department of Mathematics, ETH Zürich, Zürich, Switzerland [2]Department of Computer Science, ETH Zürich, Zürich, Switzerland [3]Department of Language Science and Technology, Saarland University, Saarbrücken, Germany. Correspondence to: Blanka Kövér <koeverb@student.ethz.ch>, Alexandra Butoi <alexandra.butoi@inf.ethz.ch>, Anej Svete <anej.svete@inf.ethz.ch>, Michael Hahn <mhahn@lst.uni-saarland.de>, Ryan Cotterell <ryan.cotterell@inf.ethz.ch>.

*Proceedings of the 43rd International Conference on Machine Learning*, Seoul, South Korea. PMLR 306, 2026. Copyright 2026 by the author(s).

| Function | $K_0(N)$ | Avg. Sens. | Learn? |
|---|---|---|---|
| PARITY | 0 | $N$ | ✗ |
| FIRST | 0 | 1 | ✗ |
| MAJORITY | $\approx 2^N$ | $\Theta(\sqrt{N})$ | ✓ (conj.) |

$K_0(N) = $ # strings with sensitivity 0

Unlearnable when $K_0(N) < N^{\frac{D-1}{2L}-5}$

*Figure 1.* Summary of our results. **Left:** Parameter space visualization. Functions with few sensitivity-0 strings (PARITY, FIRST) occupy measure-zero subsets. The MAJORITY region with positive measure is *conjectured* based on empirical observations (dashed border). Random initialization almost surely avoids measure-zero regions. **Right:** Functions with $K_0(N) < N^{(D-1)/L-5}$ are provably unlearnable; MAJORITY learnability is conjectured.

fine-grained information than the average sensitivity alone. We show (Thm. 4.5 and Cor. 4.6) that the sensitivity profile of a randomly initialized transformer has a *non-empty lower tail*: for all large enough $N$, at least polynomially many strings have low sensitivity. Equivalently, from a measure-theoretic perspective, the parameters that encode Boolean functions with very few sensitivity-zero strings occupy a Lebesgue-measure-zero subset of the parameter space. Our analysis implies that transformers cannot recognize any Boolean function which lacks such strings, a class that captures many functions of central interest to the theoretical literature on transformers—PARITY, sparse parities, and dictator functions, all of which have zero strings of sensitivity zero. Notably, our characterization rules out FIRST for long enough inputs, despite its low average sensitivity (equal to 1), a case that average-sensitivity analysis alone cannot capture. In contrast to the $\epsilon > 0$ case, our bound does *not* rule out functions like MAJORITY, whose exponentially many sensitivity-zero strings clear the polynomial floor with room to spare.

Empirically, we observe that the sensitivity profiles of randomly initialized transformers are highly skewed toward zero, corroborating our theoretical predictions. This bias persists even after training, suggesting that the inductive bias imposed by initialization constrains what functions gradient-based optimization can reach. However, we show that transformers can reliably learn MAJORITY. This leads us to conjecture that MAJORITY occupies a positive-measure subset of parameter space (Fig. 1, left)—a positive complement to the measure-zero unlearnability picture for PARITY and FIRST.

## 2. Preliminaries

Let $\mathbb{N} = \{1, 2, \dots\}$ and $\mathbb{N}_0 = \mathbb{N} \cup \{0\}$. We write $[N] \overset{\text{def}}{=} \{1, \dots, N\}$ to denote the set of natural numbers up to $N$. We use bold symbols to denote vectors and matrices. We denote by $B_p^d(r) \overset{\text{def}}{=} \{z \in \mathbb{R}^d : \|z\|_p \le r\}$ the

$d$-dimensional closed $\ell_p$-ball with radius $r$, centered at $\mathbf{0}$. $\lambda_q$ denotes the $q$-dimensional Lebesgue measure for $q \in \mathbb{N}$.

### 2.1. Boolean Functions

An **alphabet** $\Sigma$ is a finite, non-empty set of **symbols**, and a **string** $\mathbf{x}$ is a finite-length sequence of symbols. We denote by $\Sigma^N$ and $\Sigma^*$ the set of strings over $\Sigma$ of length $N$, and the set of all strings over $\Sigma$, respectively. Without loss of generality,[1] we assume that $\Sigma$ is binary, i.e., $\Sigma = \{0, 1\}$. Throughout the rest of the paper, we consider **bit strings** $\mathbf{x} \in \{0, 1\}^*$ and **Boolean functions** $f : \{0, 1\}^* \to \mathbb{R}$ mapping bit strings to real values. Unless further specified, we assume that $\mathbf{x} = x_1 \cdots x_N \in \{0, 1\}^N$ is of length $N$ where we write $x_n$ to denote $\mathbf{x}$'s $n^{\text{th}}$ bit. Additionally, we write $\mathbf{x}^{\oplus n} \in \{0, 1\}^N$ to denote the string resulting from flipping the $n^{\text{th}}$ bit $x_n$, for any $n \in [N]$.

**Language Recognition.** We now discuss language recognition by Boolean functions. Let $\Sigma$ be an alphabet. Then, a **language** $\mathcal{L} \subseteq \Sigma^*$ is a set of strings over $\Sigma$. We write $\mathcal{L}_{|N}$ for the restriction of $\mathcal{L}$ to strings of length $N$, i.e., $\mathcal{L}_{|N} \overset{\text{def}}{=} \Sigma^N \cap \mathcal{L}$.

**Definition 2.1.** *Let $\Sigma$ be an alphabet. A Boolean function $f_N : \{0, 1\}^N \to \{0, 1\}$ recognizes a language $\mathcal{L} \subseteq \Sigma^*$ if*

$$f_N(\mathbf{x}) = \begin{cases} 1, & \text{if } \mathbf{x} \in \mathcal{L}_{|N} \\ 0, & \text{if } \mathbf{x} \in \Sigma^N \setminus \mathcal{L}_{|N}. \end{cases} \tag{1}$$

*Moreover, a family of Boolean functions $\{f_N\}_{N \in \mathbb{N}_0}$ recognizes $\mathcal{L}$ if $f_N$ recognizes $\mathcal{L}$ for all $N \in \mathbb{N}_0$.*

**Sensitivity.** We recall the following definitions related to the sensitivity of Boolean functions (O'Donnell, 2014).

**Definition 2.2.** *The **sensitivity** of function $f$ on a string*

---

[1] We can always come up with a binary encoding scheme to reduce a non-binary alphabet to a binary one.

$\mathbf{x} \in \{0, 1\}^N$ is defined as

$$s_N(\mathbf{x}, f) \stackrel{\text{def}}{=} \sum_{n=1}^{N} \left| f(\mathbf{x}) - f\left(\mathbf{x}^{\oplus n}\right) \right|^2. \tag{2}$$

If $f$ maps to $\{0, 1\}$, then the sensitivity $s_N(\mathbf{x}, f)$ is the number of Hamming neighbors of $\mathbf{x}$ on which $f$ flips.

**Definition 2.3.** *The **average sensitivity** of function $f$ restricted to strings of length $N$ is defined by*

$$as_N(f) \stackrel{\text{def}}{=} \frac{1}{2^N} \sum_{\mathbf{x} \in \{0, 1\}^N} s_N(\mathbf{x}, f). \tag{3}$$

Intuitively, the average sensitivity reflects how sensitive $f$'s output is, on average, to small changes in the input. In addition to the average sensitivity, we also consider the *maximum* and *minimum sensitivity*. The maximum sensitivity measures the worst-case sensitivity of any string for a specified length $N$, while the minimum sensitivity measures the best-case sensitivity. We give formal definitions below.

**Definition 2.4.** *The **maximum sensitivity** of function $f$ restricted to strings of length $N$ is defined by*

$$maxs_N(f) \stackrel{\text{def}}{=} \max_{\mathbf{x} \in \{0, 1\}^N} s_N(\mathbf{x}, f). \tag{4}$$

**Definition 2.5.** *The **minimum sensitivity** of function $f$ restricted to strings of length $N$ is defined by*

$$mins_N(f) \stackrel{\text{def}}{=} \min_{\mathbf{x} \in \{0, 1\}^N} s_N(\mathbf{x}, f). \tag{5}$$

**Definition 2.6.** *Let $f \colon \{0, 1\}^* \to \{0, 1\}$ be a Boolean function. Then, $f$'s **sensitivity profile** (or **sensitivity spectrum**) is a collection of tuples $\{(K_0(N), K_1(N), \ldots, K_N(N))\}_{N \in \mathbb{N}_0}$ such that, for all $N$, $f$ has precisely $K_n(N)$ strings $\mathbf{x} \in \{0, 1\}^N$ with sensitivity $s_N(\mathbf{x}, f) = n$ for all $0 \le n \le N$.*

**Definition 2.7.** *Let $f \colon \{0, 1\}^N \to \mathbb{R}^P$ be a Boolean function and $\mathbf{x}$ an input string. The **absolute influence** $I_n(f, \mathbf{x})$ of the $n^{\text{th}}$ bit of $f$ for string $\mathbf{x}$ is defined as*

$$I_n(f, \mathbf{x}) \stackrel{\text{def}}{=} \left\| f(\mathbf{x}) - f(\mathbf{x}^{\oplus n}) \right\|_2. \tag{6}$$

**Examples of functions.** We study several functions that have played an important role in the existing literature on transformer expressivity and learnability.

**PARITY** evaluates to 1 if the number of 1s in the input is even, and 0 otherwise. Formally, $\text{PARITY}(\mathbf{x}) \stackrel{\text{def}}{=} 1 - x_1 \oplus \cdots \oplus x_N$, where $\oplus$ denotes the XOR operation. Because flipping any input bit changes the output, every input has sensitivity $N$, i.e., $s_N(\mathbf{x}, \text{PARITY}) = N$

for all $\mathbf{x} \in \{0, 1\}^N$. Consequently, the function's average, minimum, and maximum sensitivities all coincide with this per-input value, $as_N(\text{PARITY}) = maxs_N(\text{PARITY}) = mins_N(\text{PARITY}) = N$.

**MAJORITY** has value 1 if the input string has more 1s than 0s. Formally, $\text{MAJORITY}(\mathbf{x}) \stackrel{\text{def}}{=} \mathbb{1}\left\{\sum_{n=1}^{N} \mathbb{1}\{x_n = 1\} > \frac{N}{2}\right\}$. In MAJORITY, only the nearly balanced (with an almost equal number of 0s and 1s) inputs have nonzero sensitivity. When $N = 2m + 1$ is odd, the $2\binom{N}{m+1}$ inputs with one more ones than zeros, or vice versa, have sensitivity $m + 1$. When $N = 2m$ is even, the $\binom{N}{m}$ balanced strings have sensitivity $m$ (flipping any zero changes the output), while the $\binom{N}{m+1}$ strings with $m+1$ ones have sensitivity $m+1$ (flipping any one changes the output). All remaining (exponentially many) inputs have sensitivity 0. Taking the minimum and maximum of $s_N(\mathbf{x}', \text{MAJORITY})$ over $\mathbf{x}' \in \{0, 1\}^N$ then gives $mins_N(\text{MAJORITY}) = 0$ and $maxs_N(\text{MAJORITY}) = \lfloor N/2 \rfloor + 1$. Moreover, it can be shown that $as_N(\text{MAJORITY}) = \Theta(\sqrt{N})$ (O'Donnell, 2014, Exercise 2.22).

$m$-**SPARSE** functions (also known as $m$-juntas) depend only on at most $m$ bits of the input. Formally, a function $f \colon \{0, 1\}^N \to \{0, 1\}$ is called $m$-SPARSE if there exist $m$ indices $1 \le n_1 < \cdots < n_m \le N$ and a function $g \colon \{0, 1\}^m \to \{0, 1\}$ such that $f(x_1, \ldots, x_N) = g(x_{n_1}, \ldots, x_{n_m})$ for all $\mathbf{x} \in \{0, 1\}^N$. Examples include $m$-SPARSE PARITY and $m$-SPARSE MAJORITY functions. For a subset of $m$ indices $1 \le n_1 < \cdots < n_m \le N$, an $m$-SPARSE PARITY function $f$ is defined as $f(\mathbf{x}) \stackrel{\text{def}}{=} 1 - x_{n_1} \oplus \cdots \oplus x_{n_m}$. Similar to the PARITY function itself, such $m$-SPARSE PARITY functions have the average, minimum, and maximum sensitivity equal to $m$.

**DICTATOR functions** depend on exactly one input bit. The dictator function depending on the $n^{\text{th}}$ bit is defined as $\text{DICT}_n(\mathbf{x}) \stackrel{\text{def}}{=} \mathbb{1}\{x_n = 1\}$. The function FIRST is an instance of dictator function that depends only on the first input bit, i.e., $\text{FIRST} = \text{DICT}_1$. Dictator functions have the average, minimum, and maximum sensitivity equal to 1. Dictator functions are precisely negated 1-SPARSE PARITIES. Thus, 1-SPARSE MAJORITY is a dictator function.

## 2.2. Transformers

We study transformer encoders in the style of Hahn & Rofin (2024). We consider a binary alphabet $\Sigma = \{0, 1\}$, and an additional end-of-sequence symbol $\text{EOS} \notin \Sigma$. We restrict inputs to strings of the form $\mathbf{x}\text{EOS}$ where $\mathbf{x} \in \{0, 1\}^N$ for some $N \in \mathbb{N}_0$—a Boolean string followed by the end-of-sequence symbol. We denote such strings simply by $\mathbf{x}$, with the convention that $x_{N+1} \stackrel{\text{def}}{=} \text{EOS}$.

A **transformer** is a length-preserving function $\Sigma^*\{\text{EOS}\} \to$

$(\mathbb{R}^D)^*$. We now offer an inductive definition. To start, we assume a **symbol representation** function $\boldsymbol{e}\colon \Sigma\cup\{\text{EOS}\} \to \mathbb{R}^D$ that assigns each alphabet symbol a real vector, and a **positional encoding** function $\boldsymbol{p}\colon \mathbb{N} \to \mathbb{R}^D$ that assigns each natural number a vector. Given a string $\mathbf{x} \in \Sigma^*$, the first layer of a transformer is defined as

$$\boldsymbol{y}_n^0(\mathbf{x}) \stackrel{\text{def}}{=} \boldsymbol{e}(\mathrm{x}_n) + \boldsymbol{p}(n), \quad \forall n \in [N+1]. \qquad (7)$$

The remainder of the transformer is defined inductively. At higher layers, the activations $\boldsymbol{y}_n^\ell$ at position $n$ at the $\ell^{\text{th}}$ layer, for $\ell \in [L]$, are computed as follows. Each layer has $H$ **attention heads**. The $h^{\text{th}}$ head first computes the attention scores as follows

$$a_{nm}^{\ell h}(\mathbf{x}) \stackrel{\text{def}}{=} (\boldsymbol{K}^{\ell h}\boldsymbol{y}_m^{\ell-1}(\mathbf{x}))^\top \boldsymbol{Q}^{\ell h}\boldsymbol{y}_n^{\ell-1}(\mathbf{x}) \qquad (8a)$$

$$b_{nm}^{\ell h}(\mathbf{x}) \stackrel{\text{def}}{=} \frac{\exp a_{nm}^{\ell h}(\mathbf{x})}{\sum_{m'=1}^{N+1} \exp a_{nm'}^{\ell h}(\mathbf{x})} \qquad (8b)$$

$$\boldsymbol{c}_n^{\ell h}(\mathbf{x}) \stackrel{\text{def}}{=} \sum_{m=1}^{N+1} b_{nm}^{\ell h}(\mathbf{x})\boldsymbol{V}^{\ell h}\boldsymbol{y}_m^{\ell-1}(\mathbf{x}) \qquad (8c)$$

where $\boldsymbol{K}^{\ell h}, \boldsymbol{V}^{\ell h}, \boldsymbol{Q}^{\ell h} \in \mathbb{R}^{D\times D}$ are **key**, **query** and **value** matrices, respectively. Layers are indexed in the upper left, heads in the upper right, and positions in the lower index. Then, we construct

$$\boldsymbol{d}_n^\ell(\mathbf{x}) = \boldsymbol{y}_n^{\ell-1}(\mathbf{x}) + \nu^\ell\left(\sum_{h=1}^{H} \boldsymbol{c}_n^{\ell h}(\mathbf{x})\right) \qquad (9)$$

where $\nu^\ell\colon \mathbb{R}^D \to \mathbb{R}^D$ is a layer-specific one-layer multi-layer perceptron (MLP)

$$\nu^\ell(\boldsymbol{y}) = \boldsymbol{W}_2^\ell \operatorname{ReLU}(\boldsymbol{W}_1^\ell\boldsymbol{y} + \boldsymbol{\phi}_1^\ell) + \boldsymbol{\phi}_2^\ell \qquad (10)$$

with a residual connection, where $\boldsymbol{W}_1^\ell, \boldsymbol{W}_2^\ell \in \mathbb{R}^{D\times D}$ and $\boldsymbol{\phi}_1^\ell, \boldsymbol{\phi}_2^\ell \in \mathbb{R}^D$. Finally, we apply **layer normalization**[2] to obtain the activations. To do so, we define the following two auxiliary functions

$$\mu(\boldsymbol{v}) \stackrel{\text{def}}{=} \frac{1}{D}\sum_{d=1}^{D} v_d, \qquad (11a)$$

$$\sigma^2(\boldsymbol{v}) \stackrel{\text{def}}{=} \frac{1}{D}\|\boldsymbol{v} - \mu(\boldsymbol{v})\mathbf{1}\|_2^2 \qquad (11b)$$

for the empirical mean and variance of a vector, where $\mathbf{1} \in \mathbb{R}^D$ is the all-ones vector. We define the **normalizer** as

$$z_n^\ell(\mathbf{x}) \stackrel{\text{def}}{=} \frac{1}{\sqrt{\sigma^2(\boldsymbol{d}_n^\ell(\mathbf{x})) + \epsilon}}, \qquad (12)$$

---

[2]Transformers may differ in where exactly layer normalization is applied (Takase et al., 2023). We assume that it is applied only once per layer—specifically after the MLP block, though neither of these choices affects our results.

where $\epsilon \geq 0$ ensures numerical stability. Then, the next layer is given by

$$\boldsymbol{y}_n^\ell(\mathbf{x}) \stackrel{\text{def}}{=} z_n^\ell(\mathbf{x}) \cdot (\boldsymbol{d}_n^\ell(\mathbf{x}) - \mu(\boldsymbol{d}_n^\ell(\mathbf{x}))\mathbf{1}). \qquad (13)$$

Thus, for any string $\mathbf{x} \in \Sigma^N$, we have $\mathbf{x} \mapsto (\boldsymbol{y}_1^L(\mathbf{x}),\ldots,\boldsymbol{y}_{N+1}^L(\mathbf{x}))$, which constitutes our function.

### 2.3. Language Recognition

We now give a model of computation for the transformer. Given an $L$-layer transformer over alphabet $\Sigma$, we define a **transformer recognizer** as follows

$$T(\mathbf{x}) \stackrel{\text{def}}{=} \boldsymbol{w}^\top\boldsymbol{y}_{N+1}^L(\mathbf{x}) + \omega \qquad (14)$$

where $\boldsymbol{w} \in \mathbb{R}^D$ is a vector and $\omega \in \mathbb{R}$ a scalar. We call Eq. (14) the **read-out**. The output $T(\mathbf{x}) \in \mathbb{R}$ can be interpreted as a logit—applying the sigmoid $1/(1+e^{-z}) \in (0,1)$ produces the probability of accepting $\mathbf{x}$. We define acceptance as follows

$$\alpha_\xi(T,\mathbf{x}) \stackrel{\text{def}}{=} \begin{cases} 1 & \text{if } T(\mathbf{x}) \geq \xi \\ 0 & \text{if } T(\mathbf{x}) \leq -\xi, \\ \perp & \text{otherwise} \end{cases} \qquad (15)$$

where $\xi > 0$ is a predefined margin. Under this definition, we adopt a model of *margin-based* recognition.

**Definition 2.8.** *A transformer $T$ recognizes a family of Boolean functions $\{f_N\}_{N\in\mathbb{N}_0}$ with margin $\xi > 0$ if, for all $N \in \mathbb{N}_0$ and all $\mathbf{x} \in \{0,1\}^N$, $\alpha_\xi(T(\mathbf{x})) = f_N(\mathbf{x})$.*

Def. 2.8 is motivated by a limitation of standard (margin-less) recognition: a model may classify strings correctly yet with vanishing confidence as input length increases (Hahn, 2020; Chiang & Cholak, 2022). However, margin-based recognition comes with the consequence that—in principle—some strings may fail to be classified.

**Specialization to Boolean Functions.** With Eq. (15), the acceptance function is $\alpha_\xi(T,\cdot)\colon \{0,1\}^*\text{EOS} \to \{0,1\} \cup \{\perp\}$. When $\alpha_\xi(T,\cdot)$ classifies all strings in $\{0,1\}^*$ with margin $\xi > 0$, it is itself a Boolean function: since EOS exclusively appears at the end of input strings and is never flipped, the sensitivity definitions of §2.1 can be utilized.

### 2.4. Parameterization

A key intellectual move in our exposition will be, following Hahn (2020) and Hahn & Rofin (2024), to study a transformer recognizer as a function of its parameters. We establish the notation here.

For the remainder of the paper, we write $\Theta$ for the set of all parameters and view it as a compact subset $\Theta \subset \mathbb{R}^M$ whose

boundary $\partial\Theta$ has Lebesgue measure $0$,[3] where $M$ is the total number of scalar parameters. The transformer defined in §2.2 has the following parameters: **(i)** the per-head key, query, and value matrices $\boldsymbol{K}^{\ell h}, \boldsymbol{Q}^{\ell h}, \boldsymbol{V}^{\ell h} \in \mathbb{R}^{D \times D}$ for each layer $1 \leq \ell \leq L$ and head $1 \leq h \leq H$; **(ii)** the weights and biases $\boldsymbol{W}_1^\ell, \boldsymbol{W}_2^\ell \in \mathbb{R}^{D \times D}$, $\boldsymbol{\phi}_1^\ell, \boldsymbol{\phi}_2^\ell \in \mathbb{R}^D$ of the layer-specific one-hidden-layer MLPs for each layer $1 \leq \ell \leq L$ ; and **(iii)** the recognizer parameters $\boldsymbol{w} \in \mathbb{R}^D$ and $\omega \in \mathbb{R}$ from the readout in Eq. (14). Importantly, we *exclude* the symbol representation function and positional encodings from $\boldsymbol{\theta}$: they are often fixed (Vaswani et al., 2017), and learnable positional encodings $\boldsymbol{p} \colon \mathbb{N} \to \mathbb{R}^D$ would require infinitely many parameters, incompatible with a finite-dimensional $\Theta$ of dimension independent of $N$.

We will lift the transformer $T$ to be a function of *both* its parameters and its input string. Specifically, we write

$$T \colon \Theta \times \{0, 1\}^* \to \mathbb{R}$$
$$(\boldsymbol{\theta}, \mathbf{x}) \mapsto T(\boldsymbol{\theta}, \mathbf{x}). \tag{16}$$

It will often be useful to view a transformer $T$ as either a function of a string, given a fixed parameter vector, or as a function of its parameter vector, given a fixed string. In such cases, we write $T(\boldsymbol{\theta}, \cdot)$ and $T(\cdot, \mathbf{x})$, respectively. The same notation applies to components of the transformer, such as $\boldsymbol{d}_n^\ell$, $\boldsymbol{y}_n^\ell$, and $z_n^\ell$.

### 2.5. Blowup

For our sensitivity analyses to be carried out in §3 and §4, it is crucial to quantify the influence (Def. 2.7) of flipping a single input bit on the transformer output. Following Hahn & Rofin (2024), the central quantity of our analysis is the per-layer blowup, which measures how much the layer-norm output at layer $\ell$ can amplify a change in its input. The per-layer blowup at layer $\ell$ is

$$\tau^\ell(\boldsymbol{\theta}, \mathbf{x}) = \max_{n=1}^{N+1} \left(1 + z_n^\ell(\boldsymbol{\theta}, \mathbf{x})\right). ^4 \tag{17}$$

As we did for the transformer itself in §2.2, we lift the per-layer blowup to be a function of both the parameters and the input string: $\tau^\ell \colon \Theta \times \{0, 1\}^* \to \mathbb{R}$, with $(\boldsymbol{\theta}, \mathbf{x}) \mapsto \tau^\ell(\boldsymbol{\theta}, \mathbf{x})$. The cumulative blowup through layer $\ell$ on a string $\mathbf{x}$ is then

$$\beta^\ell(\boldsymbol{\theta}, \mathbf{x}) \overset{\text{def}}{=} \prod_{m=1}^\ell \tau^m(\boldsymbol{\theta}, \mathbf{x}). \tag{18}$$

The next lemma, adapted from Hahn & Rofin (2024), bounds the influence of flipping a single input bit on any position activation, in terms of the blowup.

---

[3]This is a technical assumption to avoid pathological counterexamples. In practice, any reasonable parameter space $\Theta$ we encounter has a measure zero boundary.

[4]We add 1 for technical reasons (Hahn & Rofin, 2024, Appendix, Lemma 16).

**Lemma 2.9.** *Let $T$ be a $D$-dimensional transformer with $L$ layers, as defined in §2.2. Fix $\epsilon \geq 0$ in Eq. (13) and $\boldsymbol{\theta} \in \Theta$. For any length $N$, any string $\mathbf{x} \in \{0, 1\}^N$, and any input positions $m, n \in [N + 1]$,*

$$\mathrm{I}_n(\boldsymbol{y}_m^L(\boldsymbol{\theta}, \cdot), \mathbf{x}) \leq C_{\text{infl}} \beta^L(\boldsymbol{\theta}, \mathbf{x}) \beta^L(\boldsymbol{\theta}, \mathbf{x}^{\oplus n}) \left(\frac{1}{N} + \delta_{m,n}\right), \tag{19}$$

*where $C_{\text{infl}}$ is a constant determined by the architecture, $\epsilon$, and $\Theta$, and $\delta_{m,n}$ is the Kronecker delta.*

## 3. Layer Norm with $\epsilon > 0$

Before delving into our measure-theoretic framework, we first analyze the simpler case where $\epsilon > 0$, which prevents the denominator of $z_n^\ell = 1/\sqrt{\sigma^2(\boldsymbol{d}_n^\ell) + \epsilon}$ from vanishing, and thus avoids an arbitrarily large blowup. We give three elementary results that, together, show that transformer recognizers are shockingly inexpressive when $\epsilon > 0$. Consequently, in §4 we consider the case where $\epsilon = 0$ to determine how much expressivity this restores.

**Stability with respect to parameters.** We first show that with $\epsilon > 0$, a transformer recognizer is Lipschitz continuous in its parameters.

**Lemma 3.1.** *Let $T$ be a $D$-dimensional transformer recognizer with $L$ layers over $\Sigma$, as defined in §2.2. Fix the layer-norm normalizer $\epsilon > 0$ in Eq. (13). Then there exists a constant $C_{\text{Lip}}$ (depending on the architecture, $\epsilon$, and $\Theta$, but not on the input string) such that, for all $\boldsymbol{\theta}_1, \boldsymbol{\theta}_2 \in \Theta$ and all $\mathbf{x} \in \Sigma^*$,*

$$|T(\boldsymbol{\theta}_1, \mathbf{x}) - T(\boldsymbol{\theta}_2, \mathbf{x})| \leq C_{\text{Lip}} \|\boldsymbol{\theta}_1 - \boldsymbol{\theta}_2\|_2. \tag{20}$$

*That is, $T$ is uniformly Lipschitz in $\boldsymbol{\theta}$ across all $\Sigma^*$.*

See App. B for a full proof. This bound has a strong consequence: the entire map from parameters to logits is Lipschitz in $\boldsymbol{\theta}$, with a constant that does not depend on the input string $\mathbf{x}$ or its length $N$. In particular, two parameter vectors that are close in $\Theta$ yield transformers whose outputs are close on *every* string, no matter how long. This input-uniform stability is what enables the packing argument of Prop. 3.5 below to count realizable functions purely in terms of the geometry of $\Theta$, with the input length entering nowhere.

**Stability with respect to inputs.** We now turn to analyzing the influence of a single input bit on the transformer output. First, we show that $\epsilon > 0$ ensures that the cumulative blowup is always bounded as follows.

**Lemma 3.2.** *Let $T$ be a $D$-dimensional transformer with $L$ layers, as defined in §2.2. If $\epsilon > 0$ in Eq. (13), then*

$$\beta^L(\boldsymbol{\theta}, \mathbf{x}) \leq \left(1 + \frac{1}{\sqrt{\epsilon}}\right)^L \tag{21}$$

*for all $\boldsymbol{\theta} \in \Theta$, $N \in \mathbb{N}_0$, and $\mathbf{x} \in \{0,1\}^N$.*

*Proof.* $\epsilon > 0$ gives $z_n^\ell = 1/\sqrt{\sigma^2(\boldsymbol{d}_n^\ell) + \epsilon} \leq 1/\sqrt{\epsilon}$ for any layer $\ell \in [L]$ and position $n \in [N+1]$. Hence

$$\tau^\ell(\boldsymbol{\theta}, \mathbf{x}) = \max_{n=1}^{N+1} \left(1 + z_n^\ell(\boldsymbol{\theta}, \mathbf{x})\right) \leq 1 + 1/\sqrt{\epsilon} \qquad (22)$$

for all $\ell \in [L]$, and

$$\beta^L(\boldsymbol{\theta}, \mathbf{x}) = \prod_{\ell=1}^L \tau^\ell(\boldsymbol{\theta}, \mathbf{x}) \leq \left(1 + \frac{1}{\sqrt{\epsilon}}\right)^L. \qquad (23)$$
∎

Recall that Lem. 2.9 provides a way to upper-bound the influence of flipping the $n^{\text{th}}$ bit in terms of the blowups $\beta^L(\boldsymbol{\theta}, \mathbf{x})$ and $\beta^L(\boldsymbol{\theta}, \mathbf{x}^{\oplus n})$. Thus for $\epsilon > 0$, it yields a $\mathcal{O}(1/N)$-bound on the change in the EOS-position activation: since we never flip the EOS position, the $\mathcal{O}(1)$ term vanishes, while Lem. 3.2 ensures that the blowup is bounded purely in terms of $L$ and $\epsilon$, independently of $N$. This is formalized in the following corollary.

**Corollary 3.3.** *Let $T$ be a $D$-dimensional transformer with $L$ layers, as defined in §2.2. Fix $\epsilon > 0$ in Eq. (13) and $\boldsymbol{\theta} \in \Theta$. For any length $N$, any string $\mathbf{x} \in \{0,1\}^N$, and any input position $n \in [N]$ (so the flipped bit is not the EOS token at position $N+1$),*

$$\mathrm{I}_n(\boldsymbol{y}_{N+1}^L(\boldsymbol{\theta}, \cdot), \mathbf{x}) = \mathcal{O}(1/N), \qquad (24)$$

*with constant depending on the architecture, $\epsilon$, and $\Theta$, but not on $N$ or $\mathbf{x}$.*

An immediate consequence of Cor. 3.3 is that any function with even a single bit that flips its output on some string cannot be recognized at margin $\xi > 0$ for sufficiently large $N$. That is, a transformer recognizer with $\epsilon > 0$ is unable to encode any non-constant Boolean function asymptotically. We make this idea formal in the following corollary.

**Corollary 3.4.** *Let $T$ be a $D$-dimensional transformer with $L$ layers, as defined in §2.2, with $\epsilon > 0$ in Eq. (13), and let $\{f_N\}_{N \in \mathbb{N}_0}$ be a family of Boolean functions. Then for every $\xi > 0$ and $\boldsymbol{\theta} \in \Theta$, $T(\boldsymbol{\theta}, \cdot)$ fails to recognize $f_N$ at margin $\xi$ for all sufficiently large $N$ such that $\mathrm{maxs}_N(f_N) \geq 1$.*

See App. C for a proof. The condition $\mathrm{maxs}_N(f_N) \geq 1$ is exactly the statement that $f_N$ is non-constant, as every non-constant Boolean function has at least one string and at least one bit whose flip changes the output. Thus, for large enough $N$, a transformer recognizer is a *constant function*.

Define the set of functions realizable at length $N$ by a transformer recognizer as follows

$$\mathcal{F}_N \overset{\text{def}}{=} \Big\{ f\colon \{0,1\}^N \to \{0,1\} \mid \exists \boldsymbol{\theta} \in \Theta:$$
$$\left(2f(\mathbf{x}) - 1\right) T(\boldsymbol{\theta}, \mathbf{x}) \geq \xi, \forall \mathbf{x} \in \{0,1\}^N \Big\}. \qquad (25)$$

Cor. 3.4 says that $|\mathcal{F}_N| \to 2$ as $N \to \infty$. However, we are also interested in non-asymptotic results. We show that a transformer recognizer with $\epsilon > 0$ can only recognize a small number of functions for all $N$. Before we state our finding, recall the notion of **diameter** of a compact set. For a non-empty, compact subset $\Theta \subset \mathbb{R}^M$, the diameter is

$$\mathrm{diam}(\Theta) \overset{\text{def}}{=} \max_{\boldsymbol{\theta}_1, \boldsymbol{\theta}_2 \in \Theta} \|\boldsymbol{\theta}_1 - \boldsymbol{\theta}_2\|_2, \qquad (26)$$

where the maximum is attained because $\|\boldsymbol{\theta}_1 - \boldsymbol{\theta}_2\|_2$ is a continuous function on the compact product $\Theta \times \Theta$.

**Proposition 3.5.** *Let $C_{Lip}$ be the uniform Lipschitz constant of Lem. 3.1. With recognition margin $\xi > 0$, let $\mathcal{F}_N$ denote the Boolean functions on $\{0,1\}^N$ recognizable in the margin-$\xi$ sense. Then for every $N \in \mathbb{N}_0$,*

$$|\mathcal{F}_N| \leq \left(1 + \frac{C_{Lip} \, \mathrm{diam}(\Theta)}{\xi}\right)^M. \qquad (27)$$

See App. D for a full proof. Note that $|\mathcal{F}_N|$ is bounded by a constant independent of $N$, while there are $2^{2^N}$ Boolean functions $\{0,1\}^N \to \{0,1\}$.

## 4. Layer Norm with $\epsilon = 0$

In §3, we analyzed the expressivity of a transformer recognizer when $\epsilon > 0$. We now investigate whether we can express a richer class of functions with $\epsilon = 0$. Interestingly, we find that while the analysis is considerably more complicated, the finding remains the same—even with $\epsilon = 0$, transformer recognizers are strikingly inexpressive.

For $\epsilon = 0$, then the blowup may in theory be arbitrarily large—no upper bound analogous to Lem. 3.2 holds deterministically. However, we find that, for any fixed string $\mathbf{x} \in \{0,1\}^N$, the blowup is still bounded *with high probability* over the choice of a random transformer not only on $\mathbf{x}$, but on a specific set of Hamming neighbors of $\mathbf{x}$.

**Definition 4.1.** *For a string $\mathbf{x} \in \{0,1\}^N$ and an index set $S \subseteq [N]$, the **Hamming neighborhood** of $\mathbf{x}$ with respect to $S$ is*

$$B_S(\mathbf{x}) \overset{\text{def}}{=} \{\mathbf{x}\} \cup \{\mathbf{x}^{\oplus n} : n \in S\}. \qquad (28)$$

We have already seen, through Lem. 2.9, that bounded blowup ensures the change in the output remains small. If the blowup is bounded on a Hamming neighborhood $B_S(\mathbf{x})$ of size $|S| = k$, then margin-based recognition implies that flipping any single bit in $S$ cannot change the output for large $N$. This means that only flipping one of the other $N - k$ bits can possibly change acceptance, so $\mathrm{s}_N(\mathbf{x}, T(\boldsymbol{\theta}, \cdot)) \leq N - k$.

Finally, we aggregate over strings and input lengths to show that almost every transformer—under the uniform

distribution on $\Theta$—has sensitivity-zero strings of length $N$ for all sufficiently large $N$. Our results also hold when the parameters $\boldsymbol{\theta}$ are drawn from certain Gaussian distributions (see App. E).

Throughout this section, constants in $\mathcal{O}$-notation may depend on the architecture and on $\Theta$, but not on the input $\mathbf{x}$ or its length $N$. Recall that $B_\infty^M(\rho) = \{\boldsymbol{z} \in \mathbb{R}^M : \|\boldsymbol{z}\|_\infty \leq \rho\}$ denotes the $\ell_\infty$-ball (hypercube) of radius $\rho$.

**The blowup is bounded with high probability.** We proceed through an intermediate step when establishing the high-probability bound, motivated by Inequality (69) of Hahn & Rofin (2024). Namely, we take a fixed transformer $T(\boldsymbol{\theta}, \cdot)$ and an input string $\mathbf{x}$, and apply a small perturbation $\boldsymbol{\Delta}$ to the parameters of $T(\boldsymbol{\theta}, \cdot)$. We prove that, with high probability over $\boldsymbol{\Delta}$, the blowup remains bounded on a Hamming neighborhood of $\mathbf{x}$ with respect to some $S$.[5]

**Lemma 4.2.** *Let $T$ be a $D$-dimensional transformer with $L$ layers, as defined in §2.2, and fix a string $\mathbf{x} \in \{0,1\}^N$. Fix $\zeta \in (2L/(D-1), 1)$.[6] Fix $\rho > 0$, and let $\boldsymbol{\Delta} \sim \mathrm{Unif}(B_\infty^M(\rho))$ be a perturbation. Then, with probability at least*

$$1 - (k+1)\left(\frac{1}{\rho}\right)^{D-1} \mathcal{O}(N^{1-(D-1)\zeta/(2L)}), \qquad (29)$$

*there exists $S \subseteq [N]$ with $|S| = k$ such that*

$$\tau^\ell(\boldsymbol{\theta} + \boldsymbol{\Delta}, \mathbf{x}') \leq 1 + N^{\zeta/(2L)} \qquad (30)$$

*for all $\mathbf{x}' \in B_S(\mathbf{x})$ and all layers $\ell \in [L]$.*

*Under these assumptions,*

$$\beta^L(\boldsymbol{\theta} + \boldsymbol{\Delta}, \mathbf{x}') = \mathcal{O}(N^{\zeta/2}) \qquad (31)$$

*for all $\mathbf{x}' \in B_S(\mathbf{x})$.*

In order to prove Lem. 4.2, we first examine how perturbing the last bias of a single layer affects the blowup at that layer, treating all previous perturbations as fixed. We capture probability through volume in parameter space: $\tau^\ell$ is bounded with high probability because the *danger zones*—parameter regions where the variance $\sigma^2(\boldsymbol{d}_n^\ell)$ is small enough to trigger a large blowup—occupy only a small volume in the parameter space. A union bound over layers $\ell \in [L]$ then gives the bound on the full blowup $\beta^L$. See App. E for a full proof.

**The output is stable with high probability.** For parameter settings where Eq. (31) holds, Lem. 2.9 allows us to bound the influence of flipping the $n^{\text{th}}$ bit of $\mathbf{x}$ for $n \in S$. This is formalized in the following corollary, which is a probabilistic analogue of Cor. 3.3 for a fixed string $\mathbf{x} \in \{0,1\}^N$.

---

[5]The probability bound depends on the size $k$ of $S$.

[6]$\zeta > 2L/(D-1)$ ensures that the probability bound tends to 1 as $N \to \infty$. Ultimately, we will choose $\zeta$ to be a number very close to 1.

**Corollary 4.3.** *Let $T$ be a $D$-dimensional transformer with $L$ layers, as defined in §2.2, and let $\mathbf{x} \in \{0,1\}^N$. Fix $\zeta \in (2L/(D-1), 1)$. Fix $\rho > 0$, and let $\boldsymbol{\Delta} \sim \mathrm{Unif}(B_\infty^M(\rho))$ be a perturbation of the transformer parameters. Then, with probability at least*

$$1 - (k+1)\left(\frac{1}{\rho}\right)^{D-1} \mathcal{O}(N^{1-(D-1)\zeta/(2L)}), \qquad (32)$$

*there exists $S \subseteq [N]$ with $|S| = k$ such that for all $n \in S$,*

$$\left|T(\boldsymbol{\theta} + \boldsymbol{\Delta}, \mathbf{x}) - T(\boldsymbol{\theta} + \boldsymbol{\Delta}, \mathbf{x}^{\oplus n})\right| = \mathcal{O}(N^{\zeta-1}). \qquad (33)$$

See App. E for a full proof.

**From perturbation to random transformer.** We now use a measure-theoretic argument to extend this bound from a neighborhood of a single parameter setting $\boldsymbol{\theta}$ to the whole parameter space $\Theta$ while holding $\mathbf{x}$ fixed. Concretely, we cover the parameter space with essentially disjoint, sufficiently small cubes so that the cover's volume closely approximates that of $\Theta$. We then apply Cor. 4.3 inside each cube, and since the probability bound holds uniformly inside each one, the bound extends to the full parameter space. See App. E for a full proof.

**Lemma 4.4.** *Let $T$ be a $D$-dimensional transformer with $L$ layers, as defined in §2.2, and let $\mathbf{x} \in \{0,1\}^N$. Let $\boldsymbol{\theta} \sim \mathrm{Unif}(\Theta)$ be a set of parameters drawn uniformly from the parameter space $\Theta \subset \mathbb{R}^M$. With probability at least*

$$1 - (k+1)\mathcal{O}(N^{1-(D-1)\zeta/(2L)}), \qquad (34)$$

*there exists $S \subseteq [N]$ with $|S| = k$ such that for all $n \in S$,*

$$\left|T(\boldsymbol{\theta}, \mathbf{x}) - T(\boldsymbol{\theta}, \mathbf{x}^{\oplus n})\right| = \mathcal{O}(N^{\zeta-1}). \qquad (35)$$

Due to margin-based recognition, Eq. (124) implies that $\mathrm{s}_N(\mathbf{x}, T(\boldsymbol{\theta}, \cdot)) \leq N - k$ with high probability if $N$ is large enough. Therefore, Lem. 4.4 establishes that any sufficiently long fixed string $\mathbf{x}$ has low sensitivity with high probability over the choice of a random transformer. An equivalent result (Lem. E.8) is shown for a Gaussian distribution in App. E.

**Transformers must have low-sensitivity strings.** We can now state our main result: *Almost every* transformer must have some low-sensitivity strings for *every* sufficiently large input length. Unlike in the results above, here we do not assume a fixed length $N$: we aggregate our result from Lem. 4.4 over all large enough $N$ using the first Borel–Cantelli lemma. See App. E for a full proof.

**Theorem 4.5.** *Let $T$ be a $D$-dimensional transformer with $L$ layers, as defined in §2.2, and let $\boldsymbol{\theta} \sim \mathrm{Unif}(\Theta)$. With probability 1, $T(\boldsymbol{\theta}, \cdot)$ has at least $\frac{1}{k+1} N^{\frac{D-1}{2L}-4}$ strings with*

*sensitivity at most $N - k$ for all sufficiently large $N$. In particular, $T(\boldsymbol{\theta}, \cdot)$ has at least $N^{\frac{D-1}{2L}-5}$ strings with sensitivity $0$ for all sufficiently large $N$.*

Thm. 4.5 is a statement about the sensitivity profile: it provides a polynomial lower bound on the number of strings with low sensitivity. By choosing $k$, we can control what constitutes low sensitivity. However, note that setting $k = N$ suffices most of the time—lower-bounding the number of sensitivity zero strings already allows us to prove useful properties about a range of formal languages.

In §3, we saw that asymptotically, every string has sensitivity zero under any transformer if $\epsilon > 0$. Thm. 4.5, in turn, states that even in the more flexible $\epsilon = 0$ case, almost every transformer has a significant number of sensitivity-zero strings for any large enough input length. Consequently, transformers asymptotically fail to recognize any language which lacks such strings: prime examples include PARITY and FIRST. We formalize this consequence in the following corollary, and discuss further applications in §5.

**Corollary 4.6.** *Let $T$ be a $D$-dimensional transformer with $L$ layers, as defined in §2.2, and let $\{f_N\}_{N \in \mathbb{N}_0}$ be a family of Boolean functions. Denote by $K_0(N)$ the number of sensitivity-zero strings of $f_N$. Then for every $\xi > 0$ and almost every $\boldsymbol{\theta} \in \Theta$, $T(\boldsymbol{\theta}, \cdot)$ fails to recognize $f_N$ at margin $\xi$ for all sufficiently large $N$ such that*

$$K_0(N) < N^{\frac{D-1}{2L}-5}. \tag{36}$$

## 5. Implications

In this section, we explore some implications of Cor. 4.6.

Eq. (36) is always satisfied when $\min s_f(N) \geq 1$. Transformers almost surely fail to recognize such $f_N$, and the reason is again a mismatch in sensitivity: $f_N$ lacks sensitivity-zero strings, but almost all transformers must have some if $N$ is large enough. We now apply this to the languages introduced in §2.

**Corollary 5.1.** *Since PARITY has minimum sensitivity $N$ for all $N$, a uniformly random transformer $T(\boldsymbol{\theta}, \cdot)$ almost surely fails to recognize it for all sufficiently large $N$.*

**Corollary 5.2.** *Since any $m$-SPARSE PARITY function has minimum sensitivity $m$ for all $N$, a uniformly random transformer $T(\boldsymbol{\theta}, \cdot)$ almost surely fails to recognize it for all sufficiently large $N$.*

**Corollary 5.3.** *Since any dictator function has minimum sensitivity $1$ for all $N$, a uniformly random transformer $T(\boldsymbol{\theta}, \cdot)$ almost surely fails to recognize it for all sufficiently large $N$.*

**Corollary 5.4.** *Since FIRST has minimum sensitivity $1$ for all $N$, a uniformly random transformer $T(\boldsymbol{\theta}, \cdot)$ almost surely fails to recognize it for all sufficiently large $N$.*

These findings do *not* contradict empirical observations of transformers successfully learning languages like FIRST, as the impossibility result applies only when $N$ is large enough. Indeed, we show that transformers learn FIRST reliably for moderate lengths (see §6, App. F). In practice, failure cases may not appear when testing on moderate-length sequences, but they necessarily exist for sufficiently long inputs.

**MAJORITY and $m$-SPARSE MAJORITIES.** Previous empirical studies indicate that transformers learn MAJORITY-type functions well. This motivates the question of whether Cor. 4.6 can be used to obtain a corresponding formal result for MAJORITY. Our theorem, however, cannot be applied in this case: MAJORITY has an exponential number of sensitivity-zero inputs, as we saw in §2.1, and thus fails to satisfy a polynomial bound like Eq. (36). The same is true for $m$-SPARSE MAJORITY functions for all $m > 1$, as can be verified through a straightforward sensitivity calculation. The result does apply to $1$-SPARSE MAJORITY functions, which correspond precisely to the dictator functions shown to be unlearnable in Cor. 5.3.

We have seen that having too few low-sensitivity strings forces unlearnability, but we do not know if the converse holds: whether an abundance of such strings guarantees learnability. Nevertheless, we conjecture that for MAJORITY, the large number of sensitivity-zero strings is reflected in a larger region of parameters that can express it.

**Conjecture 5.5.** *The set of parameter vectors $\boldsymbol{\theta} \in \Theta$ for which $T(\boldsymbol{\theta}, \cdot)$ recognizes MAJORITY with margin $\xi > 0$ has positive Lebesgue measure.*

We provide empirical support for this conjecture by showing that transformers reliably learn MAJORITY for moderate lengths (see §6, App. F).

## 6. Experiments

We empirically validate our theoretical findings by analyzing the sensitivity profiles of both randomly initialized and trained transformers. We experiment with four initialization schemes: uniform, Gaussian, Xavier uniform, and Xavier Gaussian. For each scheme, we randomly initialize a number of transformer models with randomly sampled hyperparameters and compute their sensitivity spectra. Even more strongly than predicted by Thm. 4.5, the sensitivity spectra of randomly initialized transformers are consistently heavily skewed toward zero across all initialization schemes and hyperparameter configurations. This confirms that sensitive functions occupy a vanishingly small region of parameter space, consistent with Thm. 4.5.

We additionally train transformers on PARITY, MAJORITY, FIRST, and several $m$-SPARSE PARITY, $m$-SPARSE MAJORITY, and randomly generated languages. We find

that the low-sensitivity bias persists even after training. Additionally, the sensitivity spectra show that sensitivities cluster correctly at 1 for FIRST, and at 0 and half the string length for MAJORITY. Since our unlearnability results are asymptotic, transformers can still learn FIRST at moderate input lengths. For MAJORITY, our results provide empirical evidence for Conj. 5.5. Transformers trained on PARITY fail to reach the correct sensitivity profile, corroborating our unlearnability result. Finally, for sparse parities and sparse majorities, we observe patterns that mirror those for their non-sparse counterparts. Full experimental details and results are provided in App. F.

## 7. Discussion

Our results provide a theoretical explanation for the observed bias of transformers toward low-sensitivity functions. This bias manifests not merely in terms of average sensitivity, but more fundamentally in the distributional properties of the functions they compute: for sufficiently long inputs, transformers are guaranteed to have sensitivity-zero strings. The result holds for a range of initialization schemes, demonstrating that the phenomenon is an intrinsic property of transformers.

**Is the low-sensitivity bias real?**  One might ask whether the results show a legitimate bias, or whether they merely reflect the fact that there are more low-sensitivity Boolean functions than high-sensitivity ones. The following computation shows that the former is the case. Consider a uniformly random Boolean function $f_N \colon \{0,1\}^N \to \{0,1\}$, i.e., $f_N(\mathbf{x})$ is sampled with equal probability from $\{0,1\}$, independently for each $\mathbf{x}$. Then any given input string $\mathbf{x}$ has sensitivity 0 with probability $2^{-N}$. It can be shown that the number of sensitivity-zero strings converges in distribution to a $\mathrm{Poisson}(1)$ random variable, so the probability that at least one such string exists is approximately $1 - 1/e \approx 0.63$ for large $N$. Meanwhile, our results show that a random transformer computes a function with sensitivity-zero strings with probability 1 for large $N$. This asymmetry confirms the bias toward low sensitivity.

**Implications for trained transformers.**  Our results prove the low-sensitivity bias for *randomly initialized* transformers: we show that parameter settings encoding functions with positive minimum sensitivity occupy a measure zero set. From a Bayesian perspective, this directly implies that the posterior likewise concentrates on low-sensitivity functions: any posterior is absolutely continuous with respect to the prior and therefore inherits its measure-zero sets. While training is not exactly Bayesian inference, recent work suggests a strong connection between initialization and generalization. Buzaglo et al. (2024) formalize this intuition via PAC-Bayes: the generalization capacity of a

random interpolating network *after training* is tightly linked to the volume of parameter space consistent with the target function—which our results show to be zero for high-sensitivity functions. Dziugaite & Roy (2025) strengthen the result of Buzaglo et al. (2024) further, extending it from interpolating networks to the Gibbs posterior, which softens the interpolation requirement. These results together suggest that the low-sensitivity bias we prove at initialization strongly constrains what trained transformers can generalize to: if the target function is extremely sparsely represented in parameter space, generalization toward it is very unlikely. Empirically, we also observe that this bias persists even after training (§6, App. F).

**How tight is the bound?**  Our analysis provides a *polynomial* lower bound on the number of sensitivity-zero strings as a function of input length. However, the experiments on randomly initialized transformers from §6 suggest that the true scaling might in fact be *exponential*. Proving this exponential scaling seems to require a different set of techniques, and remains an open question.

**The results are asymptotic.**  Our theoretical results predict that certain functions, such as FIRST, should be effectively unlearnable by transformers due to their low-sensitivity bias. Empirically, transformers successfully learn FIRST on practical input lengths (§6, App. F). This apparent contradiction highlights a crucial limitation of our analysis: our results are asymptotic in nature, and the threshold at which these sensitivity-zero strings necessarily appear may exceed the input lengths encountered in real-world applications.

## 8. Conclusion

We introduce a measure-theoretic framework that explains the disparity between expressivity and learnability of Boolean functions by transformers. By shifting the focus from average sensitivity to the full sensitivity profile, we generalize and strengthen prior work, proving that for any sufficiently large input length, a randomly initialized transformer almost surely has a significant number of zero-sensitivity strings. This constraint creates a fundamental learnability gap: functions that lack such strings, such as PARITY or FIRST, are provably unlearnable for long enough inputs. Experimental results across several initialization schemes corroborate our theoretical findings, confirming that sensitivity profiles of both randomly initialized and trained transformers are heavily skewed toward zero.

## Impact Statement

This paper presents work whose goal is to advance the understanding of learning capabilities of language models. There

are many potential societal consequences of our work, none of which we feel must be specifically highlighted here.

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

# Contents of the Appendix

## A. Proof of Lem. 2.9

**Lemma 2.9.** *Let $T$ be a $D$-dimensional transformer with $L$ layers, as defined in §2.2. Fix $\epsilon \geq 0$ in Eq. (13) and $\boldsymbol{\theta} \in \Theta$. For any length $N$, any string $\mathbf{x} \in \{0,1\}^N$, and any input positions $m, n \in [N+1]$,*

$$\mathrm{I}_n(\boldsymbol{y}_m^L(\boldsymbol{\theta}, \cdot), \mathbf{x}) \leq C_{\textit{infl}} \beta^L(\boldsymbol{\theta}, \mathbf{x}) \beta^L(\boldsymbol{\theta}, \mathbf{x}^{\oplus n}) \left(\frac{1}{N} + \delta_{m,n}\right), \tag{19}$$

*where $C_{\textit{infl}}$ is a constant determined by the architecture, $\epsilon$, and $\Theta$, and $\delta_{m,n}$ is the Kronecker delta.*

*Proof of Lem. 2.9.* The proof follows that of Lemma 18 in Hahn & Rofin (2024), but can be simplified in not requiring a special case for the lowest layer. We perform induction over the layer $\ell$ from 0 to $L$. First,

$$\mathrm{I}_n(\boldsymbol{y}_m^0(\boldsymbol{\theta}, \cdot), \mathbf{x}) = \mathrm{I}_n(\boldsymbol{d}_m^0(\boldsymbol{\theta}, \cdot), \mathbf{x}) = \mathcal{O}(1) \cdot \delta_{m,n} \leq C\beta^0(\boldsymbol{\theta}, \mathbf{x})\beta^0(\boldsymbol{\theta}, \mathbf{x}^{\oplus n}) \left(\frac{1}{N} + \delta_{m,n}\right) \tag{37}$$

where the $\mathcal{O}(1)$ term is bounded in terms of the magnitudes of the input layer representations, with $\beta^0$ trivially being 1 by definition.

Now for the inductive step, we show

$$\mathrm{I}_n(\boldsymbol{y}_m^\ell(\boldsymbol{\theta}, \cdot), \mathbf{x}) \leq C^{(\ell)} \cdot z_m^\ell(\boldsymbol{\theta}, \mathbf{x}) \cdot z_m^\ell(\boldsymbol{\theta}, \mathbf{x}^{\oplus n}) \cdot \left(\sum_{w=1}^N (\delta_{w,m} + \frac{1}{N})\mathrm{I}_n(\boldsymbol{y}_w^{\ell-1}(\boldsymbol{\theta}, \cdot), \mathbf{x})\right) \quad \text{(Eq 52 in Hahn \& Rofin (2024))} \tag{38}$$

$$\leq \mathcal{O}(1) \cdot \left(\prod_{j=1}^\ell C^{(j)}\right) \beta^\ell(\boldsymbol{\theta}, \mathbf{x})\beta^\ell(\boldsymbol{\theta}, \mathbf{x}^{\oplus n}) \cdot \left(\sum_{w=1}^N \sum_{v=1}^N (\delta_{w,m} + \frac{1}{N})(\delta_{v,w} + \frac{1}{N})\delta_{n,v}\right) \quad \text{(induction hypothesis)} \tag{39}$$

$$= \mathcal{O}(1) \cdot \left(\prod_{j=1}^\ell C^{(j)}\right) \beta^\ell(\boldsymbol{\theta}, \mathbf{x})\beta^\ell(\boldsymbol{\theta}, \mathbf{x}^{\oplus n}) \cdot \left(\sum_{w=1}^N (\delta_{w,m} + \frac{1}{N})(\delta_{n,w} + \frac{1}{N})\right) \tag{40}$$

$$= \mathcal{O}(1) \cdot \left(\prod_{j=1}^\ell C^{(j)}\right) \beta^\ell(\boldsymbol{\theta}, \mathbf{x})\beta^\ell(\boldsymbol{\theta}, \mathbf{x}^{\oplus n}) \cdot \left(\sum_{w=1}^N (\delta_{w,m}\delta_{n,w} + \frac{1}{N}\delta_{n,w} + \delta_{w,m}\frac{1}{N} + \frac{1}{N^2})\right) \tag{41}$$

$$= \mathcal{O}(1) \cdot \left(\prod_{j=1}^\ell C^{(j)}\right) \beta^\ell(\boldsymbol{\theta}, \mathbf{x})\beta^\ell(\boldsymbol{\theta}, \mathbf{x}^{\oplus n}) \cdot \left(\delta_{n,m} + \frac{2}{N} + \frac{1}{N^2}\right) \tag{42}$$

$$\leq \mathcal{O}(1) \left(\prod_{j=1}^\ell C^{(j)}\right) \beta^\ell(\boldsymbol{\theta}, \mathbf{x})\beta^\ell(\boldsymbol{\theta}, \mathbf{x}^{\oplus n}) \cdot \left(\delta_{n,m} + \frac{1}{N}\right) \tag{43}$$

where $C^{(\ell)}$ is a constant depending on $\Theta$ specified in Hahn & Rofin (2024), Equation (30) in terms of the norms of the parameter matrices and vectors. For our purposes, the key is that it is upper-bounded in terms of $\Theta$. ∎

## B. Proof of Lem. 3.1

**Lemma 3.1.** *Let $T$ be a $D$-dimensional transformer recognizer with $L$ layers over $\Sigma$, as defined in §2.2. Fix the layer-norm normalizer $\epsilon > 0$ in Eq. (13). Then there exists a constant $C_{\textit{Lip}}$ (depending on the architecture, $\epsilon$, and $\Theta$, but not on the input string) such that, for all $\boldsymbol{\theta}_1, \boldsymbol{\theta}_2 \in \Theta$ and all $\mathbf{x} \in \Sigma^*$,*

$$|T(\boldsymbol{\theta}_1, \mathbf{x}) - T(\boldsymbol{\theta}_2, \mathbf{x})| \leq C_{\textit{Lip}}\|\boldsymbol{\theta}_1 - \boldsymbol{\theta}_2\|_2. \tag{20}$$

*That is, $T$ is* uniformly *Lipschitz in $\boldsymbol{\theta}$ across all $\Sigma^*$.*

*Proof.* The transformer is a finite composition of the maps defined by Eqs. (7)–(13) plus the linear readout (14). We argue that each is Lipschitz in its parameters with a constant uniform in $N$; the proposition then follows from the fact that a finite composition of Lipschitz maps is Lipschitz, with constant given by the product of the individual constants. Since $e$ and $p$ are fixed, $y_n^0(\mathbf{x}) = e(\mathbf{x}_n) + p(n)$ is constant in $\theta$.

1. **Eq. (8a).** First, we turn to the attention scores. We note $a_{nm}^{\ell h}(\mathbf{x}) = (K^{\ell h} y_m^{\ell-1}(\mathbf{x}))^\top Q^{\ell h} y_n^{\ell-1}(\mathbf{x})$ is a bilinear form in $K^{\ell h}$ and $Q^{\ell h}$. Because $\Theta$ is compact, the operator norms $\|K^{\ell h}\|_{\mathrm{op}}$ and $\|Q^{\ell h}\|_{\mathrm{op}}$ are uniformly bounded above on $\Theta$, and one can show by induction on $\ell$ that the activations satisfy $\|y_n^{\ell-1}\| \leq \Gamma_{\ell-1}$ for some constants $\Gamma_{\ell-1}$, uniformly in $\mathbf{x}$. Hence the bilinear form is Lipschitz in $(K^{\ell h}, Q^{\ell h})$ with constant bounded by a product of these operator norms and $\Gamma_{\ell-1}^2$—uniformly in $N$.

2. **Eq. (8b).** Next, recall that the softmax map $\mathbb{R}^N \to \mathbb{R}^N$ is $1/4$-Lipschitz in the 2-norm (Gao & Pavel, 2018), and, importantly, $1/4$ is independent of $N$. Composing with the previous step, the attention weight $b_{nm}^{\ell h}$ is Lipschitz in $(K^{\ell h}, Q^{\ell h})$ with a constant uniform in $N$.

3. **Eq. (8c).** Next, note that $c_n^{\ell h}(\mathbf{x}) = \sum_{m=1}^N b_{nm}^{\ell h}(\mathbf{x}) V^{\ell h} y_m^{\ell-1}(\mathbf{x})$ is a convex combination (since $\sum_{m=1}^N b_{nm}^{\ell h} = 1$) of vectors of norm bounded by $\|V^{\ell h}\|_{\mathrm{op}} \cdot \Gamma_{\ell-1}$. The map is linear in $V^{\ell h}$ and Lipschitz in $(K^{\ell h}, Q^{\ell h})$ by composition with the softmax step. By compactness of $\Theta$ and the convex-combination structure, the Lipschitz constant is bounded by a product of operator norms and activation bounds—uniformly in $N$.

4. **Eq. (9).** Each $\nu^\ell$ is a feedforward network with Lipschitz activations, and its Lipschitz constant in both inputs and parameters is bounded on compact parameter sets (Virmaux & Scaman, 2018); restricting to $\Theta$ therefore gives a uniform Lipschitz constant.

5. **Eq. (13).** $y_n^\ell = z_n^\ell \cdot (d_n^\ell - \mu(d_n^\ell)\mathbf{1})$ where $z_n^\ell = 1/\sqrt{\sigma^2(d_n^\ell) + \epsilon}$. Write $u \stackrel{\mathrm{def}}{=} d_n^\ell - \mu(d_n^\ell)\mathbf{1}$. The map $d_n^\ell \mapsto u$ is a fixed linear projection and hence 1-Lipschitz. With $\epsilon > 0$ bounded away from zero, the normalizer satisfies $z_n^\ell \leq 1/\sqrt{\epsilon} < \infty$, so the multiplication $u \mapsto z_n^\ell \cdot u$ is $(1/\sqrt{\epsilon})$-Lipschitz in $u$ and $\|u\|$-Lipschitz in $z_n^\ell$. Composing, layer normalization is Lipschitz in $d_n^\ell$ (and hence in the parameters that feed it) with a constant that depends only on $\epsilon$ and the bound $\Gamma_\ell$—crucially, independent of $N$.

**Composition and the readout.** The total map $\theta \mapsto T(\theta, \mathbf{x}) = w^\top y_{N+1}^L(\theta, \mathbf{x}) + \omega$ is the composition of $L$ layers (each comprising one application of Eqs. (8a)–(13)) on top of the symbol representation (7), followed by the linear readout (14). As a function of $(w, \omega)$, the readout has gradient $(y_{N+1}^L, 1)$; its parameter-Lipschitz constant is therefore at most $\sqrt{\|y_{N+1}^L\|^2 + 1} \leq \sqrt{\Gamma_L^2 + 1}$, again $N$-independent. Each of the per-layer Lipschitz constants established above is finite and $N$-independent. Taking $C$ to be the product of these constants times the readout constant gives the bound (20) uniformly in $\mathbf{x}$. $\blacksquare$

## C. Proof of Cor. 3.4

**Corollary 3.4.** *Let $T$ be a $D$-dimensional transformer with $L$ layers, as defined in §2.2, with $\epsilon > 0$ in Eq. (13), and let $\{f_N\}_{N \in \mathbb{N}_0}$ be a family of Boolean functions. Then for every $\xi > 0$ and $\theta \in \Theta$, $T(\theta, \cdot)$ fails to recognize $f_N$ at margin $\xi$ for all sufficiently large $N$ such that $\mathrm{maxs}_N(f_N) \geq 1$.*

*Proof.* Fix any $\xi > 0$ and $\theta \in \Theta$. We show that $T(\theta, \cdot)$ fails to recognize $f_N$ at margin $\xi$ for every sufficiently large $N$ where

$$\mathrm{maxs}_N(f_N) \geq 1. \tag{44}$$

Suppose, for some such $N$, $T(\theta, \cdot)$ does recognize $f_N$ at margin $\xi$. By Eq. (44), there exist $\mathbf{x} \in \{0, 1\}^N$ and a position $n \in [N]$ with $f_N(\mathbf{x}) \neq f_N(\mathbf{x}^{\oplus n})$. Margin recognition then forces $T(\theta, \mathbf{x})$ and $T(\theta, \mathbf{x}^{\oplus n})$ to lie on opposite sides of the margin, so

$$\left| T(\theta, \mathbf{x}) - T(\theta, \mathbf{x}^{\oplus n}) \right| \geq 2\xi. \tag{45}$$

Writing $T(\boldsymbol{\theta}, \mathbf{x}) = \boldsymbol{w}^\top \boldsymbol{y}_{N+1}^L(\boldsymbol{\theta}, \mathbf{x}) + \omega$, the same difference equals $\boldsymbol{w}^\top \left( \boldsymbol{y}_{N+1}^L(\boldsymbol{\theta}, \mathbf{x}) - \boldsymbol{y}_{N+1}^L(\boldsymbol{\theta}, \mathbf{x}^{\oplus n}) \right)$, so by Cauchy–Schwarz

$$2\xi \;\leq\; \|\boldsymbol{w}\|_2 \, \|\boldsymbol{y}_{N+1}^L(\boldsymbol{\theta}, \mathbf{x}) - \boldsymbol{y}_{N+1}^L(\boldsymbol{\theta}, \mathbf{x}^{\oplus n})\|_2 \;\leq\; C(\Theta, \epsilon)/N, \tag{46}$$

where the last inequality uses Cor. 3.3 (with $C(\Theta, \epsilon)$ a constant depending on $\Theta$, $\epsilon$, and the architecture, but not on $N$) and the boundedness of $\|\boldsymbol{w}\|_2$ on $\Theta$. Rearranging gives $N \leq C(\Theta, \epsilon)/(2\xi)$. Hence, for every $N > C(\boldsymbol{\theta}, \epsilon)/(2\xi)$ at which the max-sensitivity hypothesis Eq. (44) applies, $T(\boldsymbol{\theta}, \cdot)$ fails to recognize $f_N$ at margin $\xi$. ∎

## D. Proof of Prop. 3.5

**Proposition 3.5.** *Let $C_{Lip}$ be the uniform Lipschitz constant of Lem. 3.1. With recognition margin $\xi > 0$, let $\mathcal{F}_N$ denote the Boolean functions on $\{0,1\}^N$ recognizable in the margin-$\xi$ sense. Then for every $N \in \mathbb{N}_0$,*

$$|\mathcal{F}_N| \leq \left( 1 + \frac{C_{Lip}\operatorname{diam}(\Theta)}{\xi} \right)^M. \tag{27}$$

*Proof.* For each $f \in \mathcal{F}_N$, fix a witness $\boldsymbol{\theta}_f \in \Theta$ realizing margin recognition.

*Step 1: Witnesses are separated.* Let $f_1 \neq f_2$ in $\mathcal{F}_N$, and pick $\mathbf{x}^\star \in \{0,1\}^N$ with $f_1(\mathbf{x}^\star) \neq f_2(\mathbf{x}^\star)$; without loss of generality $f_1(\mathbf{x}^\star) = 1$ and $f_2(\mathbf{x}^\star) = 0$, so margin recognition gives

$$T(\boldsymbol{\theta}_{f_1}, \mathbf{x}^\star) - T(\boldsymbol{\theta}_{f_2}, \mathbf{x}^\star) \geq 2\xi. \tag{47}$$

By Lem. 3.1, $T$ is uniformly Lipschitz in $\boldsymbol{\theta}$ across all inputs with constant $C = C_{\Theta, \epsilon}$, so Eq. (20) applied at $\mathbf{x}^\star$ yields

$$2\xi \;\leq\; |T(\boldsymbol{\theta}_{f_1}, \mathbf{x}^\star) - T(\boldsymbol{\theta}_{f_2}, \mathbf{x}^\star)| \;\leq\; C\|\boldsymbol{\theta}_{f_1} - \boldsymbol{\theta}_{f_2}\|_2, \tag{48}$$

so $\|\boldsymbol{\theta}_{f_1} - \boldsymbol{\theta}_{f_2}\|_2 \geq 2\xi/C$.

*Step 2: Disjoint balls.* Set $r = \xi/C$. Let $\mathring{B}(\boldsymbol{\theta}_f, r) \overset{\text{def}}{=} \{\boldsymbol{\theta} \in \mathbb{R}^M : \|\boldsymbol{\theta} - \boldsymbol{\theta}_f\|_2 < r\}$ denote the open ball of radius $r$ around $\boldsymbol{\theta}_f$. $\{\mathring{B}(\boldsymbol{\theta}_f, r)\}_{f \in \mathcal{F}_N}$ are pairwise disjoint: any $\boldsymbol{\theta} \in \mathring{B}(\boldsymbol{\theta}_{f_1}, r) \cap \mathring{B}(\boldsymbol{\theta}_{f_2}, r)$ would satisfy $\|\boldsymbol{\theta}_{f_1} - \boldsymbol{\theta}\|_2 < r$ and $\|\boldsymbol{\theta} - \boldsymbol{\theta}_{f_2}\|_2 < r$, so by the triangle inequality $\|\boldsymbol{\theta}_{f_1} - \boldsymbol{\theta}_{f_2}\|_2 < 2r = 2\xi/C$, contradicting Step 1. Each ball is contained in $\mathring{B}(\boldsymbol{\theta}_0, \operatorname{diam}(\Theta) + r)$ for any fixed $\boldsymbol{\theta}_0 \in \Theta$: given $\boldsymbol{\theta} \in \mathring{B}(\boldsymbol{\theta}_f, r)$, ball membership gives $\|\boldsymbol{\theta} - \boldsymbol{\theta}_f\|_2 < r$, and $\|\boldsymbol{\theta}_f - \boldsymbol{\theta}_0\|_2 \leq \operatorname{diam}(\Theta)$ since $\boldsymbol{\theta}_f, \boldsymbol{\theta}_0 \in \Theta$, so the triangle inequality yields

$$\|\boldsymbol{\theta} - \boldsymbol{\theta}_0\|_2 \;\leq\; \|\boldsymbol{\theta} - \boldsymbol{\theta}_f\|_2 + \|\boldsymbol{\theta}_f - \boldsymbol{\theta}_0\|_2 \;<\; r + \operatorname{diam}(\Theta). \tag{49}$$

Thus,

$$\bigsqcup_{f \in \mathcal{F}_N} \mathring{B}(\boldsymbol{\theta}_f, r) \subseteq \mathring{B}(\boldsymbol{\theta}_0, \operatorname{diam}(\Theta) + r). \tag{50}$$

*Step 3: Volume comparison.* Write $V_M$ for the volume of the unit ball in $\mathbb{R}^M$. The disjoint union on the left of Eq. (50) has total Lebesgue volume $|\mathcal{F}_N| \cdot V_M \, r^M$, and the containing ball on the right has volume $V_M \, (\operatorname{diam}(\Theta) + r)^M$, so

$$|\mathcal{F}_N| \cdot V_M \, r^M \;\leq\; V_M \, (\operatorname{diam}(\Theta) + r)^M. \tag{51}$$

Cancelling $V_M$ on both sides,

$$|\mathcal{F}_N| \cdot r^M \;\leq\; (\operatorname{diam}(\Theta) + r)^M. \tag{52}$$

Dividing by $r^M$ and substituting $r = \xi/C$,

$$|\mathcal{F}_N| \;\leq\; \left( \frac{\operatorname{diam}(\Theta) + r}{r} \right)^M \;=\; \left( 1 + \frac{\operatorname{diam}(\Theta)}{r} \right)^M \;=\; \left( 1 + \frac{C\operatorname{diam}(\Theta)}{\xi} \right)^M.$$

The right-hand side depends only on $\Theta$ (through $\operatorname{diam}(\Theta)$), the architecture (through $C$), and the margin $\xi$—in particular it is independent of $N$. ∎

# E. Proof of Thm. 4.5

First, let us introduce some notation. For any parameter vector $\boldsymbol{\theta} \in \mathbb{R}^M$, let $\boldsymbol{\theta}^\ell$ denote the vector containing only the parameters of layer $\ell$: the key, query, and value matrices $\boldsymbol{\theta}(\boldsymbol{K}^{\ell h}), \boldsymbol{\theta}(\boldsymbol{Q}^{\ell h}), \boldsymbol{\theta}(\boldsymbol{V}^{\ell h})$ for each head $1 \le h \le H$, as well as the weights $\boldsymbol{\theta}(\boldsymbol{W}_1^\ell), \boldsymbol{\theta}(\boldsymbol{W}_2^\ell)$ and biases $\boldsymbol{\theta}(\phi_1^\ell), \boldsymbol{\theta}(\phi_2^\ell)$ of the $\ell^{\text{th}}$ layer MLP. Let $\boldsymbol{\theta}^{\text{out}}$ denote the vector containing the recognizer parameters $\boldsymbol{\theta}(\boldsymbol{w})$ and $\boldsymbol{\theta}(\omega)$ in the readout.

## E.1. The blowup is bounded with high probability

**Low standard deviation is improbable.**    For any $\boldsymbol{z} \in \mathbb{R}^D$, recall that its (empirical) variance is $\sigma^2(\boldsymbol{z}) \stackrel{\text{def}}{=} \frac{1}{D}\|\boldsymbol{z} - \mu(\boldsymbol{z})\mathbf{1}\|_2^2$ as per Eq. (11b).

*Remark* E.1.  The variance admits a concise expression via projection. Let $U \stackrel{\text{def}}{=} \{\boldsymbol{z} \in \mathbb{R}^D \mid \langle \boldsymbol{z}, \mathbf{1} \rangle = 0\}$ be the $(D-1)$-dimensional subspace of mean-zero vectors, and let $\boldsymbol{P} \stackrel{\text{def}}{=} I_D - \frac{1}{D}\mathbf{1}\mathbf{1}^\top$ be the orthogonal projection onto $U$. Then for any $\boldsymbol{z} \in \mathbb{R}^D$,

$$\sigma^2(\boldsymbol{z}) = \frac{1}{D}\|\boldsymbol{P}\boldsymbol{z}\|_2^2. \tag{53}$$

We first investigate the effect of perturbing a fixed vector, showing that the resulting standard deviation is small only with low probability. We obtain this by expressing the probability as a ratio of volumes in the perturbation space and upper-bounding it.

**Lemma E.2.** *Let $\boldsymbol{y} \in \mathbb{R}^D$ be a vector, $\rho, \eta > 0$, and let $\boldsymbol{Z} \sim \text{Unif}(B_\infty^D(\rho))$. Then*

$$\mathbb{P}\left(\sigma^2(\boldsymbol{y} + \boldsymbol{Z}) < \eta^2\right) \le \left(\frac{\sqrt{D}\,\eta}{\rho}\right)^{D-1}. \tag{54}$$

*Proof.*  Let $\kappa \stackrel{\text{def}}{=} \sqrt{D}\eta$ and $\boldsymbol{u} \stackrel{\text{def}}{=} \boldsymbol{P}\boldsymbol{y}$, where $\boldsymbol{P}$ is defined as in Remark E.1. Then we can equivalently express

$$\sigma^2(\boldsymbol{y} + \boldsymbol{Z}) < \eta^2 \iff \frac{1}{D}\|\boldsymbol{P}(\boldsymbol{y} + \boldsymbol{Z})\|_2^2 < \eta^2 \tag{55}$$

$$\iff \|\boldsymbol{P}(\boldsymbol{y} + \boldsymbol{Z})\|_2 < \sqrt{D}\eta = \kappa \tag{56}$$

$$\iff \|\boldsymbol{P}\boldsymbol{Z} + \boldsymbol{u}\|_2 < \kappa. \tag{57}$$

We can express the probability as a ratio of volumes

$$\mathbb{P}(\|\boldsymbol{P}\boldsymbol{Z} + \boldsymbol{u}\|_2 < \kappa) = \frac{\lambda_D(\{\boldsymbol{z} \in B_\infty^D(\rho): \|\boldsymbol{P}\boldsymbol{z} + \boldsymbol{u}\|_2 < \kappa\})}{\lambda_D(B_\infty^D(\rho))}. \tag{58}$$

The vector $\mathbf{1}$ of all ones generates a line which is the orthogonal complement of $U$ as in defined in Remark E.1. Therefore, by the orthogonal decomposition $\mathbb{R}^D = U \oplus U^\perp$, each $\boldsymbol{z} \in \mathbb{R}^D$ can be uniquely decomposed as

$$\boldsymbol{z} = \boldsymbol{v} + s\mathbf{1} \quad \text{with } \boldsymbol{v} \in U,\, s \in \mathbb{R}. \tag{59}$$

Since $\boldsymbol{P}$ is the projection onto $U$, $\boldsymbol{v} = \boldsymbol{P}\boldsymbol{z}$. In order to estimate the volume in the numerator of Eq. (58), define the set

$$T(\boldsymbol{u}, \kappa) \stackrel{\text{def}}{=} \{\boldsymbol{z} \in \mathbb{R}^D: \|\boldsymbol{u} + \boldsymbol{P}\boldsymbol{z}\|_2 < \kappa\}. \tag{60}$$

One can visualize $T(\boldsymbol{u}, \kappa)$ as an infinite $D$-dimensional cylinder, along the axis given by the line $\{-\boldsymbol{u} + s\mathbf{1} \mid s \in \mathbb{R}\}$, with radius $\kappa$. Then the numerator of Eq. (58) is precisely the volume of $B_\infty^M(\rho) \cap T(\boldsymbol{u}, \kappa)$. We can express this volume as an integral over $s \in \mathbb{R}$ using Cavalieri's principle as follows. We slice $B_\infty^D(\rho) \cap T(\boldsymbol{u}, \kappa)$ by the affine hyperplanes $U + s\mathbf{1}$ for $s \in \mathbb{R}$. For each $s$, the cross-section is $\{\boldsymbol{v} \in U: \|\boldsymbol{v} + \boldsymbol{u}\|_2 < \kappa \text{ and } \|\boldsymbol{v} + s\mathbf{1}\|_\infty \le \rho\}$, and the volume equals the integral of the $(D-1)$-dimensional cross-sectional areas over $s$:

$$\lambda_D(B_\infty^D(\rho) \cap T(\boldsymbol{u}, \kappa)) = \int_{-\infty}^{\infty} \lambda_{D-1}\left(\{\boldsymbol{v} \in U: \|\boldsymbol{v} + \boldsymbol{u}\|_2 < \kappa \text{ and } \|\boldsymbol{v} + s\mathbf{1}\|_\infty \le \rho\}\right) \mathrm{d}s. \tag{61}$$

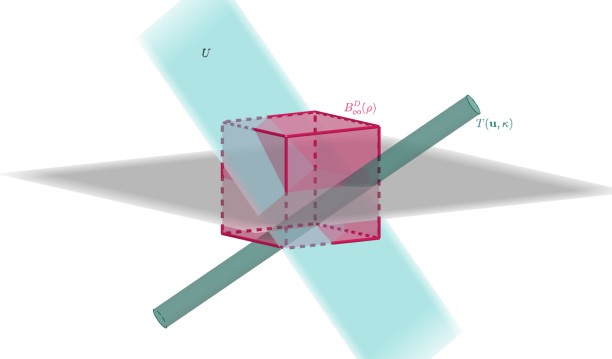

*Figure 2.* Geometric interpretation of low-variance events visualized for $D = 3$. The set $T(\boldsymbol{u}, \kappa) \cap B_\infty^D(\rho)$ captures the event of low variance, with the volume ratio given in Eq. (58).

The integrand vanishes for $|s| > \rho$: in this case, $\boldsymbol{v} + s\mathbf{1}$ falls outside the hypercube for all $\boldsymbol{v} \in U$ because $\|\boldsymbol{v} + s\mathbf{1}\|_\infty \geq \|s\mathbf{1}\|_\infty = |s| > \rho$. For $|s| \leq \rho$, dropping the constraint $\|\boldsymbol{v} + s\mathbf{1}\|_\infty \leq \rho$ only enlarges the domain of integration, giving the upper bound

$$\lambda_D(B_\infty^D(\rho) \cap T(\boldsymbol{u}, \kappa)) \leq \int_{-\rho}^{\rho} \lambda_{D-1}\left(\{\boldsymbol{v} \in U : \|\boldsymbol{v} + \boldsymbol{u}\|_2 < \kappa\}\right) \mathrm{d}s \tag{62}$$

$$= \lambda_{D-1}(B_2^{D-1}(\kappa)) \int_{-\rho}^{\rho} \mathrm{d}s \tag{63}$$

$$= 2\rho \lambda_{D-1}(B_2^{D-1}(\kappa)) \tag{64}$$

$$\leq 2\rho \lambda_{D-1}(B_2^{D-1}(\kappa)) \tag{65}$$

$$= 2\rho(2\kappa)^{D-1}. \tag{66}$$

Dividing by $\lambda_D(B_\infty^D(\rho)) = (2\rho)^D$, we obtain

$$\mathbb{P}(\|\boldsymbol{P}\boldsymbol{Z} + \boldsymbol{u}\|_2 < \kappa) \leq \frac{2\rho \, \lambda_{D-1}(B_\infty^{D-1}(\kappa))}{\lambda_D(B_\infty^D(\rho))} = \left(\frac{\kappa}{\rho}\right)^{D-1} = \left(\frac{\sqrt{D}\eta}{\rho}\right)^{D-1}, \tag{67}$$

concluding the proof. $\blacksquare$

**Layer-wise blowup bound.** For any $N \in \mathbb{N}_0$, string $\mathbf{x} \in \{0, 1\}^N$, index set $S \subseteq [N]$, and layer $\ell \in [L]$, denote by $E_{N,\mathbf{x},S,\zeta}^\ell$ the event

$$E_{N,\mathbf{x},S,\zeta}^\ell \stackrel{\text{def}}{=} \left\{\tilde{\boldsymbol{\theta}} \in \Theta : \tau^\ell(\tilde{\boldsymbol{\theta}}, \mathbf{x}') \leq 1 + N^{\zeta/(2L)} \quad \forall \mathbf{x}' \in B_S(\mathbf{x})\right\} \subset \Theta \tag{68}$$

that the blowup at layer $\ell$ is bounded as $N^{\zeta/(2L)}$ on $\mathbf{x}$ and on $\mathbf{x}^{\oplus n}$ for all $n \in S$ simultaneously.

We now examine the effect of randomly perturbing the last bias of layer $\ell$, given that all preceding perturbations are fixed. Applying Lem. E.2, the variance of $\boldsymbol{d}_n^\ell$ is small only with low probability, so the normalizers $z_n^\ell$ and the blowup $\tau^\ell$ at this layer are bounded with high probability.

**Definition E.3.** *For a perturbation vector* $\boldsymbol{\Delta} = \left[(\boldsymbol{\Delta}^1)^\top, \ldots, (\boldsymbol{\Delta}^L)^\top, (\boldsymbol{\Delta}^{out})^\top\right]^\top$, *denote by*

$$\boldsymbol{\Delta}^{<\ell} \stackrel{\text{def}}{=} \left[(\boldsymbol{\Delta}^1)^\top, \ldots, (\boldsymbol{\Delta}^{\ell-1})^\top, \boldsymbol{\Delta}(\boldsymbol{K}^{\ell h})^\top, \cdots, \boldsymbol{\Delta}(\boldsymbol{W}_2^\ell)^\top, \mathbf{0}^\top, \mathbf{0}^\top, \ldots, \mathbf{0}^\top\right]^\top \tag{69}$$

*the vector only containing the perturbations to layers up to* $\ell - 1$ *and to the* $\ell^{th}$ *layer excluding the last bias, and by*

$$\boldsymbol{\Delta}^{\leq\ell} \stackrel{\text{def}}{=} \left[(\boldsymbol{\Delta}^1)^\top, \ldots, (\boldsymbol{\Delta}^{\ell-1})^\top, \boldsymbol{\Delta}(\boldsymbol{K}^{\ell h})^\top, \cdots, \boldsymbol{\Delta}(\boldsymbol{W}_2^\ell)^\top, \boldsymbol{\Delta}(\boldsymbol{\phi}_2^\ell)^\top, \mathbf{0}^\top, \ldots, \mathbf{0}^\top\right]^\top \tag{70}$$

*the vector containing the perturbations to the entire first $\ell$ layers. The same notation applies to fixed realizations $\boldsymbol{\delta}$.*

**Lemma E.4.** *Fix $\boldsymbol{\theta} \in \Theta$, and let $T(\boldsymbol{\theta}, \cdot)$ be a $D$-dimensional transformer with $L$ layers, as defined in §2.2. Further, let $\mathbf{x} \in \{0, 1\}^N$, and $S \subseteq [N]$ an index set with $|S| = k$.*

*Fix a layer $\ell \in [L]$, fix a realization $\boldsymbol{\Delta}^{<\ell} = \boldsymbol{\delta}^{<\ell}$, and draw the remaining perturbations $\boldsymbol{\Delta}(\phi_2^\ell), \boldsymbol{\Delta}^{\ell+1}, \ldots, \boldsymbol{\Delta}^L, \boldsymbol{\Delta}^{out}$ uniformly at random. Then*

$$\mathbb{P}\left(\boldsymbol{\theta} + \boldsymbol{\Delta} \in E_{N,\mathbf{x},S,\zeta}^\ell \mid \boldsymbol{\Delta}^{<\ell} = \boldsymbol{\delta}^{<\ell}\right) \geq 1 - (k+1)\left(\frac{1}{\rho}\right)^{D-1} \mathcal{O}(N^{1-(D-1)\zeta/(2L)}). \tag{71}$$

*Proof of Lem. E.4.* Consider $\boldsymbol{d}_n^\ell(\boldsymbol{\theta} + \boldsymbol{\Delta}^{<\ell}, \tilde{\mathbf{x}})$ and $\boldsymbol{d}_n^\ell(\boldsymbol{\theta} + \boldsymbol{\Delta}^{\leq\ell}, \tilde{\mathbf{x}})$, as defined in Eq. (9), at layer $\ell$ on an arbitrary input string $\tilde{\mathbf{x}}$, at any position $n$. Since we have fixed the value of every perturbation preceding $\boldsymbol{\Delta}(\phi_2^\ell)$, $\boldsymbol{d}_n^\ell(\boldsymbol{\theta} + \boldsymbol{\Delta}^{<\ell}, \tilde{\mathbf{x}})$ is fixed while $\boldsymbol{d}_n^\ell(\boldsymbol{\theta} + \boldsymbol{\Delta}^{\leq\ell}, \tilde{\mathbf{x}})$ is random: specifically, Eq. (9) and Eq. (10) together show that

$$\boldsymbol{d}_n^\ell(\boldsymbol{\theta} + \boldsymbol{\Delta}^{\leq\ell}, \tilde{\mathbf{x}}) = \boldsymbol{d}_n^\ell(\boldsymbol{\theta} + \boldsymbol{\Delta}^{<\ell}, \tilde{\mathbf{x}}) + \boldsymbol{\Delta}(\phi_2^\ell). \tag{72}$$

Note that $\boldsymbol{\Delta}^{\ell+1}, \ldots, \boldsymbol{\Delta}^L, \boldsymbol{\Delta}^{out}$ do not affect $\boldsymbol{d}_n^\ell(\boldsymbol{\theta} + \boldsymbol{\Delta}^{\leq\ell}, \tilde{\mathbf{x}})$, and the random coordinates of $\boldsymbol{\Delta}$ are independent. Hence we can directly invoke Lem. E.2 with $\boldsymbol{y} \stackrel{\text{def}}{=} \boldsymbol{d}_n^\ell(\boldsymbol{\theta} + \boldsymbol{\Delta}^{<\ell}, \tilde{\mathbf{x}})$, $\eta \stackrel{\text{def}}{=} N^{-\zeta/(2L)}$, and $\boldsymbol{Z} \stackrel{\text{def}}{=} \boldsymbol{\Delta}(\phi_2^\ell)$ to obtain that

$$\mathbb{P}\left(\sigma^2(\boldsymbol{d}_n^\ell(\boldsymbol{\theta} + \boldsymbol{\Delta}^{\leq\ell}, \tilde{\mathbf{x}})) < N^{-\zeta/L} \mid \boldsymbol{\Delta}^{<\ell} = \boldsymbol{\delta}^{<\ell}\right) \tag{73}$$

$$= \mathbb{P}\left(\sigma^2(\boldsymbol{d}_n^\ell(\boldsymbol{\theta} + \boldsymbol{\Delta}^{<\ell}, \tilde{\mathbf{x}}) + \boldsymbol{\Delta}(\phi_2^\ell)) < N^{-\zeta/L} \mid \boldsymbol{\Delta}^{<\ell} = \boldsymbol{\delta}^{<\ell}\right) \quad \text{(Eq. (72))} \tag{74}$$

$$\leq \left(\frac{\sqrt{D}N^{-\zeta/(2L)}}{\rho}\right)^{D-1} \quad \text{(Lem. E.2)} \tag{75}$$

for any fixed $\tilde{\mathbf{x}}$ and any position $n$. A union bound over all $n \in [N+1]$ gives

$$\mathbb{P}\left(\exists n \in [N+1]: \sigma^2(\boldsymbol{d}_n^\ell(\boldsymbol{\theta} + \boldsymbol{\Delta}^{\leq\ell}, \tilde{\mathbf{x}})) < N^{-\zeta/L} \mid \boldsymbol{\Delta}^{<\ell} = \boldsymbol{\delta}^{<\ell}\right) \tag{76}$$

$$\leq (N+1)\left(\frac{\sqrt{D}N^{-\zeta/(2L)}}{\rho}\right)^{D-1} = \left(\frac{1}{\rho}\right)^{D-1} \mathcal{O}(N^{1-(D-1)\zeta/(2L)}) \tag{77}$$

where the factor depending only on $D$ is absorbed into $\mathcal{O}(\cdot)$. For any string $\tilde{\mathbf{x}}$, Eq. (76) provides an upper bound on the probability that the variance is close to zero at some position. Applying this to each $\mathbf{x}' \in B_S(\mathbf{x})$ separately therefore gives the lower bound

$$\mathbb{P}\left(\forall \mathbf{x}' \in B_S(\mathbf{x}), \forall n \in [N+1] \quad \sigma^2(\boldsymbol{d}_n^\ell(\boldsymbol{\theta} + \boldsymbol{\Delta}^{\leq\ell}, \mathbf{x}')) \geq N^{-\zeta/L} \mid \boldsymbol{\Delta}^{<\ell} = \boldsymbol{\delta}^{<\ell}\right) \tag{78}$$

$$= 1 - \mathbb{P}\left(\exists \mathbf{x}' \in B_S(\mathbf{x}), \exists n \in [N+1]: \sigma^2(\boldsymbol{d}_n^\ell(\boldsymbol{\theta} + \boldsymbol{\Delta}^{\leq\ell}, \mathbf{x}')) \geq N^{-\zeta/L} \mid \boldsymbol{\Delta}^{<\ell} = \boldsymbol{\delta}^{<\ell}\right) \quad \text{(complement event)} \tag{79}$$

$$\geq 1 - \sum_{\mathbf{x}' \in B_S(\mathbf{x})} \mathbb{P}\left(\exists n \in [N+1]: \sigma^2(\boldsymbol{d}_n^\ell(\boldsymbol{\theta} + \boldsymbol{\Delta}^{\leq\ell}, \mathbf{x}')) < N^{-\zeta/L} \mid \boldsymbol{\Delta}^{<\ell} = \boldsymbol{\delta}^{<\ell}\right) \quad \text{(union bound)} \tag{80}$$

$$\geq 1 - (k+1)\left(\frac{1}{\rho}\right)^{D-1} \mathcal{O}(N^{1-(D-1)\zeta/(2L)}). \quad \text{(Eq. (76) for } \tilde{\mathbf{x}} \stackrel{\text{def}}{=} \mathbf{x}') \tag{81}$$

It remains to show that if $\boldsymbol{\Delta}^{<\ell} = \boldsymbol{\delta}^{<\ell}$ and $\sigma^2(\boldsymbol{d}_n^\ell(\boldsymbol{\theta} + \boldsymbol{\Delta}^{\leq\ell}, \mathbf{x}')) \geq N^{-\zeta/L}$ for all $\mathbf{x}' \in B_S(\mathbf{x})$ and all $n \in [N+1]$, then $\boldsymbol{\theta} + \boldsymbol{\Delta}^{\leq\ell} \in E_{N,\mathbf{x},S,\zeta}^\ell$. For such $\boldsymbol{\Delta}$,

$$z_n^\ell(\boldsymbol{\theta} + \boldsymbol{\Delta}^{\leq\ell}, \mathbf{x}') = \frac{1}{\sqrt{\sigma^2(\boldsymbol{d}_n^\ell(\boldsymbol{\theta} + \boldsymbol{\Delta}^{\leq\ell}, \mathbf{x}')) + \epsilon}} \leq \frac{1}{\sqrt{\sigma^2(\boldsymbol{d}_n^\ell(\boldsymbol{\theta} + \boldsymbol{\Delta}^{\leq\ell}, \mathbf{x}'))}} \leq N^{\zeta/(2L)} \tag{82}$$

holds for all $\mathbf{x}' \in B_S(\mathbf{x})$ and all $n \in [N+1]$. Hence

$$\tau^\ell(\boldsymbol{\theta} + \boldsymbol{\Delta}^{\leq\ell}, \mathbf{x}') = \max_{n \in [N+1]}\{1 + z_n^\ell(\boldsymbol{\theta} + \boldsymbol{\Delta}^{\leq\ell}, \mathbf{x}')\} \leq 1 + N^{\zeta/(2L)} \tag{83}$$

for all $\mathbf{x}' \in B_S(\mathbf{x})$, which is precisely $\boldsymbol{\theta} + \boldsymbol{\Delta}^{\leq\ell} \in E_{N,\mathbf{x},S,\zeta}^\ell$. ∎

*Remark* E.5. The proof goes through for any $\epsilon \geq 0$. For $\epsilon > 0$, the normalization factor $z_n^\ell$ is deterministically upper-bounded by $1/\sqrt{\epsilon}$, independent of $N$, which is stronger than the probabilistic bound derived here. The probabilistic argument is therefore only necessary in the $\epsilon = 0$ case.

**Full blowup bound.** We prove the following stronger statement. Then the claim we need follows as a special case.

**Lemma E.6.** *Let $T$ be a $D$-dimensional transformer with $L$ layers, as defined in §2.2. Let $\mathbf{x} \in \{0, 1\}^N$, and let $S \subseteq [N]$ with $|S| = k$ be a fixed index set. Fix $\zeta \in (2L/(D-1), 1)$. Fix $\rho > 0$, and let $\boldsymbol{\Delta} \sim \text{Unif}(B_\infty^M(\rho))$ be a perturbation. Then, with probability at least $1 - (k+1)\left(\frac{1}{\rho}\right)^{D-1}\mathcal{O}(N^{1-(D-1)\zeta/(2L)})$,*

$$\tau^\ell(\boldsymbol{\theta} + \boldsymbol{\Delta}, \mathbf{x}') \leq 1 + N^{\zeta/(2L)} \tag{84}$$

*for all $\mathbf{x}' \in B_S(\mathbf{x})$ and all $\ell \in [L]$.*

*Proof of Lem. E.6.* Using the notation from Eq. (68), our goal is to show that

$$\boldsymbol{\theta} + \boldsymbol{\Delta} \in \bigcap_{\ell \in [L]} E_{N,\mathbf{x},S,\zeta}^\ell \tag{85}$$

with probability at least $1 - (k+1)\left(\frac{1}{\rho}\right)^{D-1}\mathcal{O}(N^{1-(D-1)\zeta/(2L)})$.

We first show that, for any fixed $\ell \in [L]$, we can drop the conditioning and still maintain the bound: i.e., that Lem. E.4 implies

$$\mathbb{P}\left(\boldsymbol{\theta} + \boldsymbol{\Delta} \in E_{N,\mathbf{x},S,\zeta}^\ell\right) \geq 1 - (k+1)\left(\frac{1}{\rho}\right)^{D-1}\mathcal{O}(N^{1-(D-1)\zeta/(2L)}). \tag{86}$$

Let $\mathbb{P}$ denote the probability measure associated with the uniform distribution on the hypercube. Then we can express the left-hand side, using the law of total probability, as

$$\mathbb{P}\left(\boldsymbol{\theta} + \boldsymbol{\Delta} \in E_{N,\mathbf{x},S,\zeta}^\ell\right) = \mathbb{P}\left(\boldsymbol{\theta} + \boldsymbol{\Delta}^{\leq \ell} \in E_{N,\mathbf{x},S,\zeta}^\ell\right) \tag{87}$$

$$= \int_{\boldsymbol{\delta}^{<\ell}} \left(\int_{\boldsymbol{\delta}(\phi_2^\ell)} \mathbb{1}\left\{\boldsymbol{\theta} + \boldsymbol{\delta}^{\leq \ell} \in E_{N,\mathbf{x},S,\zeta}^\ell\right\} d\mathbb{P}(\boldsymbol{\delta}(\phi_2^\ell))\right) d\mathbb{P}(\boldsymbol{\delta}^{<\ell}) \tag{88}$$

$$= \int_{\boldsymbol{\delta}^{<\ell}} \mathbb{P}\left(\boldsymbol{\theta} + \boldsymbol{\delta}^{\leq \ell} \in E_{N,\mathbf{x},S,\zeta}^\ell \mid \boldsymbol{\Delta}^{<\ell} = \boldsymbol{\delta}^{<\ell}\right) d\mathbb{P}(\boldsymbol{\delta}^{<\ell}) \tag{89}$$

$$\geq 1 - (k+1)\left(\frac{1}{\rho}\right)^{D-1}\mathcal{O}(N^{1-(D-1)\zeta/(2L)}), \quad \text{(Lem. E.4)} \tag{90}$$

since we integrate with respect to a probability measure. Finally, a union bound over $\ell \in [L]$ gives

$$\mathbb{P}\left(\boldsymbol{\theta} + \boldsymbol{\Delta} \in \bigcap_{\ell \in [L]} E_{N,\mathbf{x},S,\zeta}^\ell\right) = 1 - \mathbb{P}\left(\exists \ell \in [L] : \boldsymbol{\theta} + \boldsymbol{\Delta} \notin E_{N,\mathbf{x},S,\zeta}^\ell\right) \tag{91}$$

$$\geq 1 - \sum_{\ell \in [L]} \mathbb{P}\left(\boldsymbol{\theta} + \boldsymbol{\Delta} \notin E_{N,\mathbf{x},S,\zeta}^\ell\right) \geq 1 - L(k+1)\left(\frac{1}{\rho}\right)^{D-1}\mathcal{O}(N^{1-(D-1)\zeta/(2L)}) \quad \text{(Eq. (86))} \tag{92}$$

$$= 1 - (k+1)\left(\frac{1}{\rho}\right)^{D-1}\mathcal{O}(N^{1-(D-1)\zeta/(2L)}). \quad \text{($L$ absorbed into $\mathcal{O}(\cdot)$)} \tag{93}$$

∎

**Lemma 4.2.** *Let $T$ be a $D$-dimensional transformer with $L$ layers, as defined in §2.2, and fix a string $\mathbf{x} \in \{0,1\}^N$. Fix $\zeta \in (2L/(D-1), 1)$.[7] Fix $\rho > 0$, and let $\mathbf{\Delta} \sim \mathrm{Unif}(B_\infty^M(\rho))$ be a perturbation. Then, with probability at least*

$$1 - (k+1)\left(\frac{1}{\rho}\right)^{D-1} \mathcal{O}(N^{1-(D-1)\zeta/(2L)}), \tag{29}$$

*there exists $S \subseteq [N]$ with $|S| = k$ such that*

$$\tau^\ell(\boldsymbol{\theta} + \mathbf{\Delta}, \mathbf{x}') \leq 1 + N^{\zeta/(2L)} \tag{30}$$

*for all $\mathbf{x}' \in B_S(\mathbf{x})$ and all layers $\ell \in [L]$.*

*Under these assumptions,*

$$\beta^L(\boldsymbol{\theta} + \mathbf{\Delta}, \mathbf{x}') = \mathcal{O}(N^{\zeta/2}) \tag{31}$$

*for all $\mathbf{x}' \in B_S(\mathbf{x})$.*

*Proof of Lem. 4.2.* Immediate from Lem. E.6, since the bound holds for any fixed $S$. ∎

### E.2. The output is stable with high probability

**Corollary 4.3.** *Let $T$ be a $D$-dimensional transformer with $L$ layers, as defined in §2.2, and let $\mathbf{x} \in \{0,1\}^N$. Fix $\zeta \in (2L/(D-1), 1)$. Fix $\rho > 0$, and let $\mathbf{\Delta} \sim \mathrm{Unif}(B_\infty^M(\rho))$ be a perturbation of the transformer parameters. Then, with probability at least*

$$1 - (k+1)\left(\frac{1}{\rho}\right)^{D-1} \mathcal{O}(N^{1-(D-1)\zeta/(2L)}), \tag{32}$$

*there exists $S \subseteq [N]$ with $|S| = k$ such that for all $n \in S$,*

$$\left|T(\boldsymbol{\theta} + \mathbf{\Delta}, \mathbf{x}) - T(\boldsymbol{\theta} + \mathbf{\Delta}, \mathbf{x}^{\oplus n})\right| = \mathcal{O}(N^{\zeta-1}). \tag{33}$$

*Proof of Cor. 4.3.* Lem. 4.2 shows that with probability at least $1 - (k+1)\left(\frac{1}{\rho}\right)^{D-1} \mathcal{O}(N^{1-(D-1)\zeta/(2L)})$, there exists a set $S \subseteq [N]$ of $k$ indices such that $\boldsymbol{\theta} + \mathbf{\Delta} \in \bigcap_{\ell \in [L]} E_{N,\mathbf{x},S,\zeta}^\ell$.

Let $\mathbf{\Delta}$ be such that $\boldsymbol{\theta} + \mathbf{\Delta} \in \bigcap_{\ell \in [L]} E_{N,\mathbf{x},S,\zeta}^\ell$. Since each per-layer blowup is bounded as $\tau^\ell(\boldsymbol{\theta} + \mathbf{\Delta}, \mathbf{x}') \leq 1 + N^{\zeta/(2L)}$ on all $\mathbf{x}' \in B_S(\mathbf{x})$, the overall blowup is bounded as

$$\beta^L(\boldsymbol{\theta} + \mathbf{\Delta}, \mathbf{x}') \overset{\mathrm{def}}{=} \prod_{\ell=1}^L \tau^\ell(\boldsymbol{\theta} + \mathbf{\Delta}, \mathbf{x}') \leq (1 + N^{\zeta/(2L)})^L = \mathcal{O}(N^{\zeta/2}). \tag{94}$$

Lem. 2.9 then shows that

$$\mathrm{I}_n(\boldsymbol{y}_m^L, \mathbf{x}) \leq C\beta^L(\boldsymbol{\theta} + \mathbf{\Delta}, \mathbf{x})\beta^L(\boldsymbol{\theta} + \mathbf{\Delta}, \mathbf{x}^{\oplus n})\left(\frac{1}{N} + \delta_{m,n}\right) = \mathcal{O}(N^\zeta)\left(\frac{1}{N} + \delta_{m,n}\right) \tag{95}$$

for any $n \in S$ and $m \in [N+1]$. In particular,

$$\mathrm{I}_n(\boldsymbol{y}_{N+1}^L, \mathbf{x}) = \left\|\boldsymbol{y}_{N+1}^L(\mathbf{x}) - \boldsymbol{y}_{N+1}^L(\mathbf{x}^{\oplus n})\right\|_2 = \mathcal{O}(N^{\zeta-1}), \tag{96}$$

---

[7] $\zeta > 2L/(D-1)$ ensures that the probability bound tends to 1 as $N \to \infty$. Ultimately, we will choose $\zeta$ to be a number very close to 1.

since we never flip the EOS token, and the Kronecker delta term vanishes. Finally, since the readout layer is Lipschitz, this yields

$$\left|T(\boldsymbol{\theta}+\boldsymbol{\Delta},\mathbf{x})-T(\boldsymbol{\theta}+\boldsymbol{\Delta},\mathbf{x}^{\oplus n})\right| = \left|\boldsymbol{w}^\top(\boldsymbol{y}_{N+1}^L(\mathbf{x})-\boldsymbol{y}_{N+1}^L(\mathbf{x}^{\oplus n}))\right| \tag{97}$$

$$\leq \|\boldsymbol{w}\|_2 \left\|\boldsymbol{y}_{N+1}^L(\mathbf{x})-\boldsymbol{y}_{N+1}^L(\mathbf{x}^{\oplus n}))\right\|_2 = \mathcal{O}(N^{\zeta-1}) \tag{98}$$

as we wanted.

■

### E.3. From perturbation to random transformer

**Lemma 4.4.** *Let $T$ be a $D$-dimensional transformer with $L$ layers, as defined in §2.2, and let $\mathbf{x} \in \{0,1\}^N$. Let $\boldsymbol{\theta} \sim \mathrm{Unif}(\Theta)$ be a set of parameters drawn uniformly from the parameter space $\Theta \subset \mathbb{R}^M$. With probability at least*

$$1 - (k+1)\mathcal{O}(N^{1-(D-1)\zeta/(2L)}), \tag{34}$$

*there exists $S \subseteq [N]$ with $|S| = k$ such that for all $n \in S$,*

$$\left|T(\boldsymbol{\theta},\mathbf{x})-T(\boldsymbol{\theta},\mathbf{x}^{\oplus n})\right| = \mathcal{O}(N^{\zeta-1}). \tag{35}$$

*Proof of Lem. 4.4.* Let

$$Q_{N,\mathbf{x},k,\zeta} \stackrel{\text{def}}{=} \left\{\tilde{\boldsymbol{\theta}} \in \mathbb{R}^M : \exists S \subseteq [N], \, |S| = k : |T(\tilde{\boldsymbol{\theta}},\mathbf{x})-T(\tilde{\boldsymbol{\theta}},\mathbf{x}^{\oplus n})| = \mathcal{O}(N^{\zeta-1}) \quad \forall \mathbf{x}' \in B_S(\mathbf{x})\right\} \subset \mathbb{R}^M \tag{99}$$

denote the event that the difference in logits is bounded as $\mathcal{O}(N^{\zeta-1})$ on a Hamming neighborhood of size $k$ of $N$. Our goal is to show that

$$\mathbb{P}(\boldsymbol{\theta} \in Q_{N,\mathbf{x},k,\zeta}) = \frac{\lambda_M(Q_{N,\mathbf{x},k,\zeta} \cap \Theta)}{\lambda_M(\Theta)} \geq 1 - (k+1)\mathcal{O}(N^{1-(D-1)\zeta/(2L)}). \tag{100}$$

We first construct a $\chi$-approximating cover $\Theta_\chi \supset \Theta$ for each $\chi > 0$ such that

$$\lambda_M(\Theta_\chi) < \lambda_M(\Theta) + \chi \tag{101}$$

and

$$\frac{\lambda_M(Q_{N,\mathbf{x},k,\zeta} \cap \Theta_\chi)}{\lambda_M(\Theta_\chi)} \geq 1 - (k+1)\mathcal{O}(N^{1-(D-1)\zeta/(2L)}). \tag{102}$$

Then we show that this implies Eq. (100) by taking an infimum over all $\chi > 0$.

**Constructing $\Theta_\chi$.** Let us fix $\chi > 0$. We construct the cover $\Theta_\chi$ as an essentially disjoint union of small cubes. For any $r \in \mathbb{N}$,[8] "partition" $\mathbb{R}^M$ into a grid of cubes $C_{\boldsymbol{a}}$, $\boldsymbol{a} = (a_1, \ldots, a_M) \in \mathbb{Z}^M$ of side length $1/r$, where

$$C_{\boldsymbol{a}}^r \stackrel{\text{def}}{=} \bigtimes_{i=1}^M \left[\frac{a_i}{r}, \frac{a_i+1}{r}\right]. \tag{103}$$

Let $\mathcal{C}^r \stackrel{\text{def}}{=} \{C_{\boldsymbol{a}}^r : \boldsymbol{a} \in \mathbb{Z}^M\}$. The elements of $\mathcal{C}^r$ do not form a partition in the strict sense since they are not disjoint. However, they are *essentially* disjoint: they only intersect on a measure zero set, which is negligible for our purposes.

Denote by

$$\mathcal{C}^r(\Theta) \stackrel{\text{def}}{=} \{C_{\boldsymbol{a}}^r \in \mathcal{C}^r : C_{\boldsymbol{a}}^r \cap \Theta \neq \emptyset\} \tag{104}$$

---

[8]We will choose a suitable value of $r$ later.

the set of cubes which intersect $\Theta$, and their union by

$$\text{Cover}^{(r)} \stackrel{\text{def}}{=} \bigcup_{C_a^r \in \mathcal{C}^r(\Theta)} C_a^r. \tag{105}$$

It is clear that $\text{Cover}^{(r)} \supset \Theta$, hence $\lambda_M(\Theta) \leq \lambda_M\left(\text{Cover}^{(r)}\right)$ by monotonicity. Furthermore, since $\Theta$ is bounded and its boundary is a null set, $\Theta$ is Jordan measurable. Jordan measurability ensures that $\lambda_M\left(\text{Cover}^{(r)}\right) \to \lambda_M(\Theta)$ as $r \to \infty$, so there exists a large enough integer $r$ such that

$$\lambda_M\left(\text{Cover}^{(r)}\right) - \lambda_M(\Theta) < \chi. \tag{106}$$

Fix now this choice of $r$, and let $\Theta_\chi \stackrel{\text{def}}{=} \text{Cover}^{(r)}$. Condition Eq. (101) is given precisely by Eq. (106), while Eq. (102) holds because Cor. 4.3 holds in each small cube independently. Note that size of the cubes in this partition is determined by $\Theta$, hence we can treat $\rho$ as a constant and absorb the term $\left(\frac{1}{\rho}\right)^{D-1}$ in $\mathcal{O}(\cdot)$.

**Proving Eq. (100).** Now we turn to showing how Eq. (101) and Eq. (102) for all $\chi > 0$ imply Eq. (100).

First, note that

$$\frac{\lambda_M(Q_{N,\mathbf{x},k,\varsigma} \cap \Theta_\chi)}{\lambda_M(\Theta_\chi)} = 1 - \frac{\lambda_M(\Theta_\chi \setminus Q_{N,\mathbf{x},k,\varsigma})}{\lambda_M(\Theta_\chi)}, \tag{107}$$

and Eq. (102) is thus equivalent to

$$\frac{\lambda_M(Q_{N,\mathbf{x},k,\varsigma} \cap \Theta_\chi)}{\lambda_M(\Theta_\chi)} \geq 1 - (k+1)\mathcal{O}(N^{1-(D-1)\varsigma/(2L)}) \iff \frac{\lambda_M(\Theta_\chi \setminus Q_{N,\mathbf{x},k,\varsigma})}{\lambda_M(\Theta_\chi)} \leq (k+1)\mathcal{O}(N^{1-(D-1)\varsigma/(2L)}). \tag{108}$$

We can then bound

$$\mathbb{P}(\boldsymbol{\theta} \in Q_{N,\mathbf{x},k,\varsigma}) = \frac{\lambda_M(Q_{N,\mathbf{x},k,\varsigma} \cap \Theta)}{\lambda_M(\Theta)} = \frac{\lambda_M(\Theta) - \lambda_M(\Theta \setminus Q_{N,\mathbf{x},k,\varsigma})}{\lambda_M(\Theta)} \tag{109}$$

$$\geq 1 - \frac{\lambda_M(\Theta_\chi \setminus Q_{N,\mathbf{x},k,\varsigma})}{\lambda_M(\Theta)} \quad (\Theta_\chi \supset \Theta) \tag{110}$$

$$\geq 1 - \frac{\lambda_M(\Theta_\chi)(k+1)\mathcal{O}(N^{1-(D-1)\varsigma/(2L)})}{\lambda_M(\Theta)} \quad (\text{Eq. (102)}) \tag{111}$$

$$\geq 1 - \frac{(\lambda_M(\Theta) + \chi)(k+1)\mathcal{O}(N^{1-(D-1)\varsigma/(2L)})}{\lambda_M(\Theta)} \quad (\text{Eq. (101)}) \tag{112}$$

$$= 1 - \left(1 + \frac{\chi}{\lambda_M(\Theta)}\right)(k+1)\mathcal{O}(N^{1-(D-1)\varsigma/(2L)}). \tag{113}$$

Taking the infimum over all $\chi > 0$, we obtain the desired bound Eq. (100). ■

### E.4. Transformers must have low-sensitivity strings

**Theorem 4.5.** *Let $T$ be a $D$-dimensional transformer with $L$ layers, as defined in §2.2, and let $\boldsymbol{\theta} \sim \text{Unif}(\Theta)$. With probability 1, $T(\boldsymbol{\theta}, \cdot)$ has at least $\frac{1}{k+1} N^{\frac{D-1}{2L}-4}$ strings with sensitivity at most $N - k$ for all sufficiently large $N$. In particular, $T(\boldsymbol{\theta}, \cdot)$ has at least $N^{\frac{D-1}{2L}-5}$ strings with sensitivity $0$ for all sufficiently large $N$.*

*Proof of Thm. 4.5.* Let $C$ be the constant of $\mathcal{O}(N^{\varsigma-1})$ in Lem. 4.4, and let $N_0$ denote the index such that $CN^{\varsigma-1} < 2\xi$ for all $N \geq N_0$. The existence of $N_0$ is guaranteed since $\varsigma < 1$ implies that $N^{\varsigma-1} \to 0$ as $N \to \infty$. Then for $N \geq N_0$, $|T(\boldsymbol{\theta}, \mathbf{x}) - T(\boldsymbol{\theta}, \mathbf{x}^{\oplus n})| \leq CN^{\varsigma-1} < 2\xi$ implies that $\alpha_\xi(T(\boldsymbol{\theta}, \cdot), \mathbf{x}) = \alpha_\xi(T(\boldsymbol{\theta}, \cdot), \mathbf{x}^{\oplus n})$.

We first show that for any subset of strings $\mathcal{X} \subseteq \{0,1\}^N$, $|\mathcal{X}| = K$ where $N \geq N_0$, with high probability all strings $\mathbf{x} \in \mathcal{X}$ have sensitivity $s_N(\mathbf{x}, T(\boldsymbol{\theta}, \cdot))) \leq N - k$. Indeed,

$$\mathbb{P}\left(\bigcap_{\mathbf{x}\in\mathcal{X}}\{s_N(\mathbf{x},T(\boldsymbol{\theta},\cdot)))\leq N-k\}\right)=1-\mathbb{P}\left(\bigcup_{\mathbf{x}\in\mathcal{X}}\{s_N(\mathbf{x},T(\boldsymbol{\theta},\cdot)))>N-k\}\right) \tag{114}$$

$$\geq 1-\sum_{\mathbf{x}\in\mathcal{X}}\mathbb{P}(s_N(\mathbf{x},T(\boldsymbol{\theta},\cdot)))>N-k) \quad \text{(union bound)} \tag{115}$$

$$\geq 1-K\cdot(k+1)\mathcal{O}(N^{1-(D-1)\zeta/(2L)}). \quad \text{(Lem. 4.4)} \tag{116}$$

We now use the first Borel–Cantelli lemma (Durrett, 2010, Theorem 2.3.1.) to show that, for suitable choices of the function $K(N)$, almost surely there exist $K(N)$ strings $\mathbf{x} \in \{0,1\}^N$ with sensitivity $s_N(\mathbf{x}, T(\boldsymbol{\theta}, \cdot)) \leq N - k$ for all large enough $N$.

**Lemma E.7** (Borel–Cantelli). *Let $\mathcal{A}_1, \mathcal{A}_2, \ldots$ be a sequence of events in a probability space. If $\sum_{N=1}^{\infty} \mathbb{P}(\mathcal{A}_N) < \infty$, then almost surely only finitely many of the events occur.*

Let $K: \mathbb{N}_0 \to \mathbb{R}^+$. Let $\mathcal{B}_N$ denote the event that there exists a subset of strings $\mathcal{X} \subseteq \{0,1\}^N$, $|\mathcal{X}| = K(N)$ such that every $\mathbf{x} \in \mathcal{X}$ has sensitivity $s_N(\mathbf{x}, T(\boldsymbol{\theta}, \cdot)) \leq N - k$, and let $\mathcal{A}_N \stackrel{\text{def}}{=} \mathcal{B}_N^{\mathsf{c}}$ denote its complement. Then for $N \geq N_0$, we can bound

$$\mathbb{P}(\mathcal{B}_N)=\mathbb{P}\left(\bigcup_{\substack{\mathcal{X}\subseteq\{0,1\}^N\\|\mathcal{X}|=K(N)}}\bigcap_{\mathbf{x}\in\mathcal{X}}\{s_N(\mathbf{x},T(\boldsymbol{\theta},\cdot))\leq N-k\}\right)\geq 1-K(N)\cdot(k+1)\mathcal{O}(N^{1-(D-1)\zeta/(2L)}) \tag{117}$$

using Eq. (114), or equivalently, for the complement,

$$\mathbb{P}(\mathcal{A}_N)\leq K(N)\cdot(k+1)\mathcal{O}(N^{1-(D-1)\zeta/(2L)}). \tag{118}$$

Thus we can write

$$\sum_{N=1}^{\infty}\mathbb{P}(\mathcal{A}_N)=C'+\sum_{N=N_0}^{\infty}\mathbb{P}(\mathcal{A}_N) \tag{119}$$

where we aggregated the first $N_0 - 1$ terms into a (finite) constant $C'$. We need to guarantee that the sum

$$\sum_{N=N_0}^{\infty}\mathbb{P}(\mathcal{A}_N)\leq\sum_{N=N_0}^{\infty}K(N)\cdot(k+1)\mathcal{O}(N^{1-(D-1)\zeta/(2L)})\stackrel{?}{<}\infty \tag{120}$$

is finite, in order to satisfy the conditions of Borel–Cantelli. A condition that suffices is

$$K(N)(k+1)\mathcal{O}(N^{1-(D-1)\zeta/(2L)})=\mathcal{O}(N^{-1-\varsigma}) \tag{121}$$

for some $\varsigma > 0$, which corresponds to

$$K(N)=\mathcal{O}(\frac{1}{k+1}N^{-2+(D-1)\zeta/(2L)-\varsigma}). \tag{122}$$

Since we can take $\zeta < 1$ to be arbitrarily close to 1 and $\varsigma > 0$ arbitrarily close to 0, Eq. (122) is implied if

$$K(N)=\mathcal{O}(\frac{1}{k+1}N^{-4+(D-1)/(2L)}). \tag{123}$$

Consider, in particular, the function $K(N) \stackrel{\text{def}}{=} \frac{1}{k+1}N^{-4+(D-1)/(2L)}$. In light of the above discussion, we can conclude by Borel–Cantelli that almost surely only finitely many of the events $\mathcal{A}_N$'s occur for this choice of $K(N)$. In other words, almost surely there exist $\frac{1}{k+1}N^{-4+(D-1)/(2L)}$ strings with sensitivity $s_N(\mathbf{x}, T(\boldsymbol{\theta}, \cdot)) \leq N - k$ for all sufficiently large $N$, which was what we wanted to show. ∎

### E.5. Proof assuming Gaussian measure

Here, we show a version of Lemma 4.4 for a Gaussian measure over the parameters, instead of the uniform measure over a compact set. The rest of the proof remains identical for the Gaussian case.

**Lemma E.8.** *There is $\gamma > 0$ such that the following holds. Let $T$ be a $D$-dimensional transformer with $L$ layers, as defined in §2.2, and let $\mathbf{x} \in \{0,1\}^N$. Let $\boldsymbol{\theta} \sim \mathcal{N}(\mathbf{0}, \frac{1}{\gamma D}\boldsymbol{I}_M)$ be a set of parameters drawn from a Gaussian distribution with mean $\mathbf{0}$ and covariance $\frac{1}{\gamma D}$ times the identity. With probability at least $1 - (k+1)\mathcal{O}(N^{1-(D-1)\zeta/(2L)})$, there exists $S \subseteq [N]$ with $|S| = k$ such that for all $n \in S$,*

$$\left|T(\boldsymbol{\theta}, \mathbf{x}) - T(\boldsymbol{\theta}, \mathbf{x}^{\oplus n})\right| = \mathcal{O}(N^{\zeta-1}). \tag{124}$$

*Here the $\mathcal{O}(\cdot)$ terms include constants depending on the architecture, but not $N$ or $\mathbf{x}$.*

*Remark* E.9. We believe that Lem. E.8 can be strengthened to hold for $\gamma = 1$, which would extend the result to Xavier Gaussian initializations.

*Proof of Lem. E.8.* Let $\mathbb{P}$ be the measure associated with the Gaussian prior for $\boldsymbol{\theta}$.

As in the proof of Lem. 4.4, let

$$Q_{N,\mathbf{x},k,\zeta} \overset{\text{def}}{=} \left\{\tilde{\boldsymbol{\theta}} \in \mathbb{R}^M : \exists S \subseteq [N], |S| = k : |T(\tilde{\boldsymbol{\theta}}, \mathbf{x}) - T(\tilde{\boldsymbol{\theta}}, \mathbf{x}^{\oplus n})| = \mathcal{O}(N^{\zeta-1}) \quad \forall \mathbf{x}' \in B_S(\mathbf{x})\right\} \subset \mathbb{R}^M \tag{125}$$

denote the event that the difference in logits is bounded as $\mathcal{O}(N^{\zeta-1})$ on a Hamming neighborhood of size $k$ of $N$.

We construct cubes $C_{\boldsymbol{a}}^r$ as in the proof of Lem. 4.4. $\boldsymbol{a} = (a_1, \dots, a_M) \in \mathbb{Z}^M$ of side length $1/r$, where

$$C_{\boldsymbol{a}}^r \overset{\text{def}}{=} \bigtimes_{i=1}^M \left[\frac{a_i}{r}, \frac{a_i + 1}{r}\right]. \tag{126}$$

We can apply Cor. 4.3 separately to each $C_{\boldsymbol{a}}^r$. In each case, the $\mathcal{O}(\cdot)$ implies a constant $Const(C_{\boldsymbol{a}}^r)$ upper-bounded by $\sup_{\boldsymbol{\theta} \in C_{\boldsymbol{a}}^r} \mathcal{O}(\text{poly}(\|\boldsymbol{\theta}\|, M) \cdot \exp(4M \cdot \|\boldsymbol{\theta}\|^2))$. $Const(C_{\boldsymbol{a}}^r)$ arises because the $\mathcal{O}(\cdot)$ in Cor. 4.3 hides a constant that depends on the norms of the vectors in $\Theta$. This poses no issue when covering a compact space with finitely many such $C_{\boldsymbol{a}}^r$'s, but becomes critical for an infinite number of such $C_{\boldsymbol{a}}^r$'s in the Gaussian case. In this latter scenario, one must account for how this hidden constant depends on $C_{\boldsymbol{a}}^r$. Namely, the constant is $C^{(1)} \cdots C^{(L)}$ in Lemma 18 of Hahn & Rofin (2024), which by Appendix B.1 in Hahn & Rofin (2024) is bounded by quantities polynomial in $M$ and $\|\boldsymbol{\theta}\|$, times $\exp(4D \sum_\ell \max_h \|(\boldsymbol{K}^{\ell h})^\top \boldsymbol{Q}^{\ell h}\|_{\text{spectral}}) \leq \exp(4M \sum_\ell \max_h \|(\boldsymbol{K}^{\ell h})^\top \boldsymbol{Q}^{\ell h}\|_{\text{spectral}})$. Given that the spectral norm is upper-bounded by the Frobenius norm, this is upper-bounded by $\exp(4M \sum_\ell \max_h \|\boldsymbol{K}^{\ell h}\|_F \|\boldsymbol{Q}^{\ell h}\|_F) \leq \exp(4M\|\boldsymbol{\theta}\|_2^2)$ where we used that the $\boldsymbol{K}^{\ell h}$ and $\boldsymbol{Q}^{\ell h}$ matrices are disjoint slices of the vector $\boldsymbol{\theta}$.

In fact, the choice of $r$ will play no role; we can take it to be 1. Thus $\left(\frac{1}{\rho}\right)^{D-1} = 1$ in Eq. (32). We denote the cubes simply as $C_{\boldsymbol{a}}$ from now on.

We take $\gamma$ large enough ($\gamma = \frac{20M}{D}$) to make $\sigma^2 = \frac{1}{\gamma D} = \frac{1}{20M}$.

Consider

$$\mathbb{P}(\boldsymbol{\theta} \notin Q_{N,\mathbf{x},k,\zeta}) = \sum_{\boldsymbol{a}} \mathbb{P}(\boldsymbol{\theta} \in C_{\boldsymbol{a}})\mathbb{P}(\boldsymbol{\theta} \notin Q_{N,\mathbf{x},k,\zeta} \mid \boldsymbol{\theta} \in C_{\boldsymbol{a}}) \tag{127}$$

$$\leq \sum_{\boldsymbol{a}} \mathbb{P}(\boldsymbol{\theta} \in C_{\boldsymbol{a}})(k+1)Const(C_{\boldsymbol{a}})N^{1-(D-1)\zeta/(2L)} \tag{128}$$

$$= N^{1-(D-1)\zeta/(2L)}(k+1)\sum_{\boldsymbol{a}} \mathbb{P}(\boldsymbol{\theta} \in C_{\boldsymbol{a}})Const(C_{\boldsymbol{a}}) \tag{129}$$

The sum is by definition independent of $N$; all that remains is to show that it is also finite:

$$\sum_{\boldsymbol{a}} \mathbb{P}(\boldsymbol{\theta} \in C_{\boldsymbol{a}}) Const(C_{\boldsymbol{a}}) \tag{130}$$

$$\leq \sum_{\boldsymbol{a}} \mathbb{P}(\boldsymbol{\theta} \in C_{\boldsymbol{a}}) \sup_{\boldsymbol{\theta} \in C_{\boldsymbol{a}}} \mathcal{O}(\text{poly}(\|\boldsymbol{\theta}\|, D) \cdot \exp(4M \cdot \|\boldsymbol{\theta}\|^2)) \tag{131}$$

$$\leq \sum_{\boldsymbol{a}} \left( \sup_{\boldsymbol{\theta} \in C_{\boldsymbol{a}}} \mathbb{P}(\boldsymbol{\theta}) \right) \lambda_M(C_{\boldsymbol{a}}) \sup_{\boldsymbol{\theta} \in C_{\boldsymbol{a}}} \mathcal{O}(\text{poly}(\|\boldsymbol{\theta}\|, D) \cdot \exp(4M \cdot \|\boldsymbol{\theta}\|^2)) \tag{132}$$

$$= \frac{1}{\sqrt{(2\pi)}^M \sigma^M} \sum_{\boldsymbol{a}} \exp\left( -\frac{\sum_{i=1}^M a_i^2}{\sigma^2} \right) \sup_{\boldsymbol{\theta} \in C_{\boldsymbol{a}}} \mathcal{O}(\text{poly}(\|\boldsymbol{\theta}\|, D) \cdot \exp(4M \cdot \|\boldsymbol{\theta}\|^2)) \tag{133}$$

$$\leq \frac{1}{\sqrt{(2\pi)}^M \sigma^M} \sum_{\boldsymbol{a}} \exp\left( -\frac{\sum_{i=1}^M a_i^2}{\sigma^2} \right) \mathcal{O}(\text{poly}(\|\boldsymbol{a}\|, D) \cdot \exp(8 \cdot M \cdot \|\boldsymbol{a}\|^2)) \tag{134}$$

$$\leq \frac{1}{\sqrt{(2\pi)}^M \sigma^M} \sum_{\boldsymbol{a}} \text{poly}(\|\boldsymbol{a}\|, D) \cdot \exp\left( -\frac{\|\boldsymbol{a}\|_2^2}{2\sigma^2} + 8 \cdot M \cdot \|\boldsymbol{a}\|^2 \right) \tag{135}$$

$$= \frac{1}{\sqrt{(2\pi)}^M \sigma^M} \sum_{\boldsymbol{a}} \text{poly}(\|\boldsymbol{a}\|, D) \cdot \exp\left( \|\boldsymbol{a}\|_2^2 \left[ \frac{-1}{2\sigma^2} + 8 \cdot M \right] \right) \tag{136}$$

$$\leq \frac{1}{\sqrt{(2\pi)}^M \sigma^M} \sum_{\boldsymbol{a}} \text{poly}(\|\boldsymbol{a}\|, D) \cdot \exp\left( \|\boldsymbol{a}\|_2^2 \left[ \frac{-20M}{2} + 8 \cdot M \right] \right) \tag{137}$$

$$\leq \frac{1}{\sqrt{(2\pi)}^M \sigma^M} \sum_{\boldsymbol{a}} \text{poly}(\|\boldsymbol{a}\|, D) \cdot \exp\left( -2M\|\boldsymbol{a}\|_2^2 \right) \tag{138}$$

where $\mathbb{P}(\boldsymbol{\theta})$ also denotes the Gaussian density (overloading notation). By comparing the last sum to integrals of the form $\int_{\mathbb{R}^M} \|\boldsymbol{a}\|^p \cdot \exp(-\|\boldsymbol{a}\|_2^2) d\boldsymbol{a}$ for $p \in \mathbb{N}_0$, we find that the sum is finite. As the sum is finite, we can absorb it into $\mathcal{O}(\cdot)$.

∎

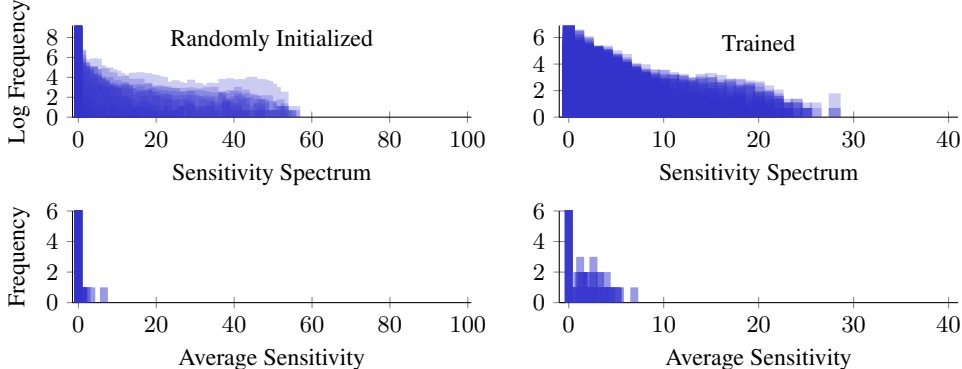

*Figure 3.* **Top**: Sensitivity spectra of 500 (50 trials × 10 datasets) randomly initialized transformers with uniform weights vs. trained transformers initialized using Xavier normal. Frequencies are shown on a log scale to improve visibility. **Bottom**: Distribution of average sensitivities of 500 (50 trials × 10 datasets) randomly initialized transformers with uniform weights vs. trained transformers initialized using Xavier normal. The $y$-axis is clipped to visualize better the tail of higher-sensitivity outliers.

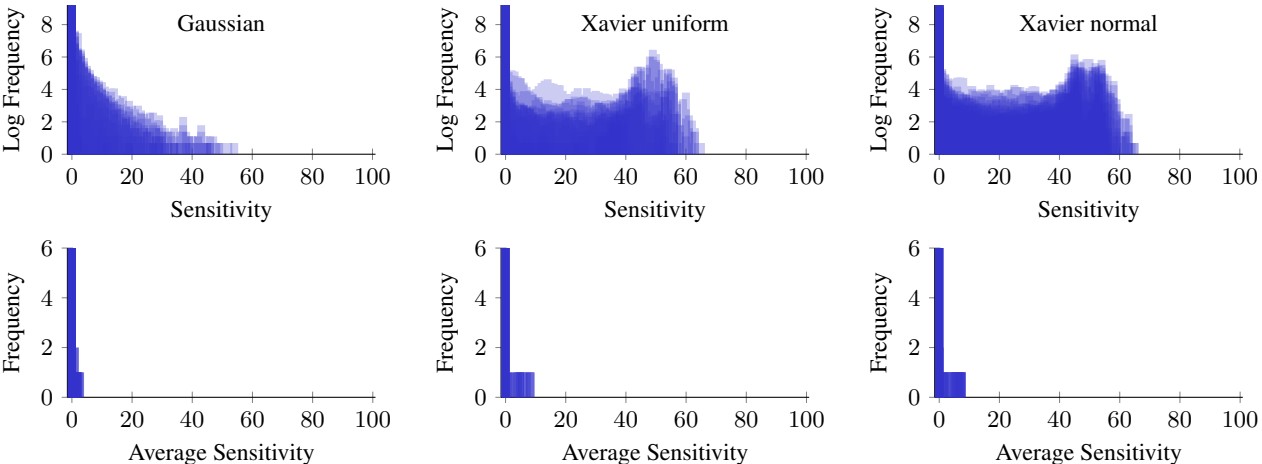

*Figure 4.* **Top**: Sensitivity spectra of 500 (50 trials × 10 datasets) randomly initialized transformers with Gaussian weights, Xavier uniform, and Xavier normal, respectively. Frequencies are shown on a log scale to improve visibility. **Bottom**: Distribution of average sensitivities of 500 (50 trials × 10 datasets) randomly initialized transformers with Gaussian weights, Xavier uniform, and Xavier normal, respectively. The $y$-axis is clipped to visualize the tail of higher-sensitivity outliers.

# F. Experiments

Here, we give further details about the experimental design and the full results summarized in §6.

## F.1. Randomly initialized transformers

For the randomly initialized models, we experiment with four parameter (weights and biases) initialization schemes: uniform, Gaussian, Xavier uniform, and Xavier Gaussian (Glorot & Bengio, 2010). For uniform and Gaussian initialization, parameters are drawn from $[-1, 1]$ and $\mathcal{N}(0, 1)$, respectively; for both Xavier uniform and Gaussian, weights are sampled with a gain of 1. We use fixed positional encodings, namely sinusoidal positional encodings. For each initialization method, we sample 10 datasets containing $10k$ strings. We use strings of length 100. For each initialization scheme and dataset, we randomly initialize 50 transformers, and we compute the sensitivity spectrum and the average sensitivity. For each of these transformers, we randomly sample the hyperparameters, and use a small constant for $\epsilon$. We sample the number of layers uniformly from $[2, 6]$. We sample the number of attention heads from $\{1, 2, 4, 8\}$ and the head size from $\{16, 32, 64, 128\}$, then we set the model dimension as their product. The feedforward dimension is set as a multiple of the model dimension, where the multiplier is drawn uniformly from $\{2, 4, 8\}$. Fig. 3 (Left) shows the sensitivity spectra and average sensitivities for the uniform initialization. In Fig. 4, we show the sensitivity spectra and average sensitivities of transformers initialized

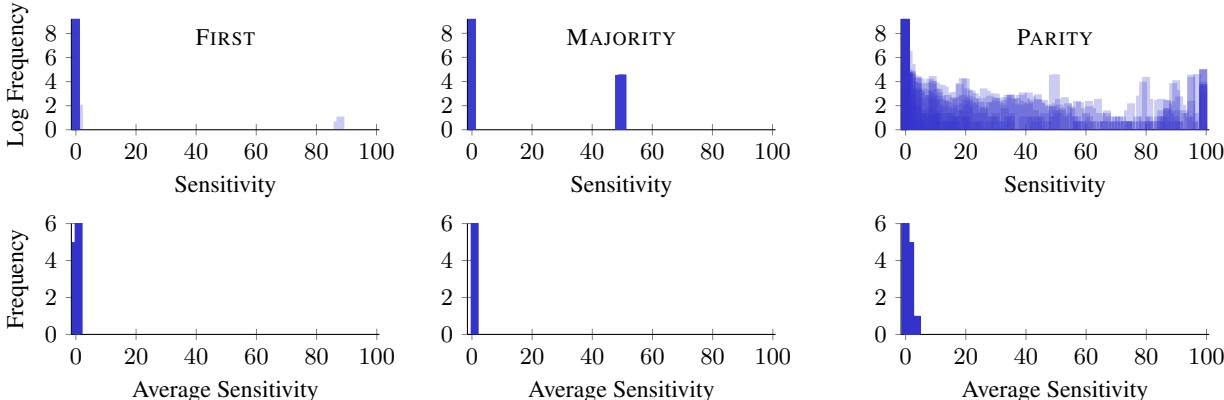

*Figure 5.* **Top**: Sensitivity spectra of 50 transformers (per language) after training to convergence on the languages FIRST, PARITY, and MAJORITY. The parameters of all models are initialized uniformly. Frequencies are shown on a log scale to improve visibility. **Bottom**: Distribution of average sensitivities of 50 transformers (per language) after training to convergence on the languages FIRST, PARITY, and MAJORITY. The parameters of all models are initialized uniformly. The $y$-axis is clipped to visualize better the higher-sensitivity outliers.

using the other three methods. The results empirically validate our theoretical framework. As predicted by Thm. 4.5, the sensitivity spectra (Fig. 3, Top left) reveal that the vast majority of strings have sensitivity 0 or close to 0. While prior work (Bhattamishra et al., 2023; Hahn & Rofin, 2024) characterized this bias using average sensitivity, our analysis offers a finer-grained view. The tight clustering of average sensitivities near zero (Fig. 3, Bottom left) confirms that high average-sensitivity transformers occupy a vanishingly small volume of the parameter space.

### F.2. Trained transformers

We train transformers on the languages PARITY, MAJORITY, FIRST, and several $m$-SPARSE PARITY, $m$-SPARSE MA-JORITY, and randomly generated languages. For each sparse variant, we generate 10 distinct languages for each value of $m \in \{5, 10, 20, 50\}$. In all experiments, we use strings of length 100, training sets of size $10k$, and validation and test sets of size $1k$. For random languages, we use strings of length 40, and training sets of size $10k$. The random languages are generated by sampling uniformly each bit of the string, as well as the label. We train 10 different transformer models initialized with uniform weights on each language. The hyperparameters, including model dimensions, are randomly sampled as described for the random transformers. We initialize the models trained on the random languages using Xavier normal, as opposed to uniformly, because we observed that this aids learning in preliminary experiments. We randomly sample the batch size from a uniform distribution over $[128, 4096]$, and the initial learning rate from a log-uniform distribution over $[0.0001, 0.01]$.

We train each model by optimizing the binary cross entropy between the true label and the prediction using Adam (Kingma & Ba, 2015). We take a checkpoint every epoch, where we evaluate the model on the validation set and update the learning rate and early stopping schedules. We multiply the learning rate by 0.5 after 5 epochs of no decrease and stop early after 10 epochs of no decrease. On the randomly sampled languages, we evaluate on the training set instead, as done by Bhattamishra et al. (2023).

We show in Fig. 5, Fig. 6, and Fig. 7 the sensitivity spectra and the average sensitivities of the models trained on PARITY, MAJORITY, and FIRST, and the sparse parities and majorities, respectively. All models trained on FIRST and MAJORITY learn the languages perfectly. Indeed, the sensitivity spectra show that sensitives cluster correctly at 1 for FIRST, and at 0 and half the string length for MAJORITY. On PARITY, the trained models have higher-sensitivity strings. We show in Fig. 3 (Right) the sensitivity spectra and average sensitivities of trained transformers on randomly generated languages. Some of the sampled hyperparameter configurations lead to models that cannot learn these languages. For these models, the (average) sensitivities cluster around 0.

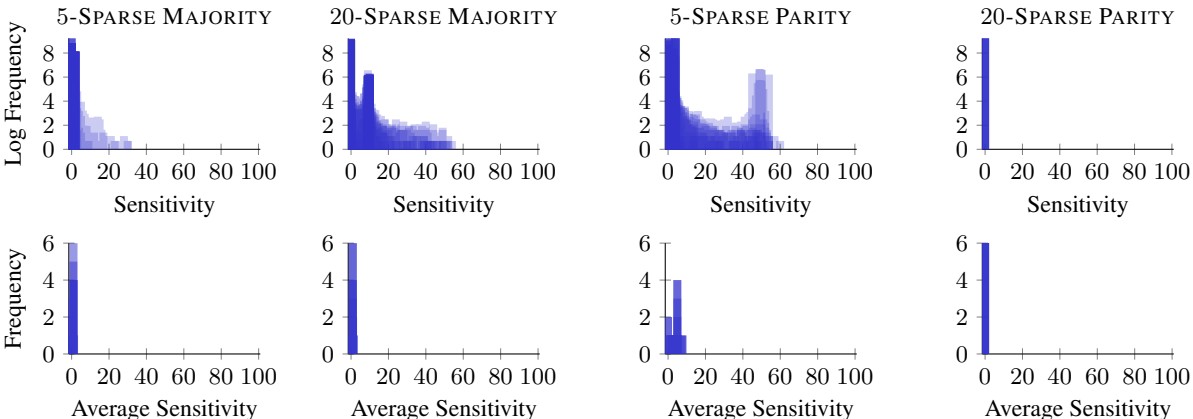

*Figure 6.* **Top**: Sensitivity spectra of 50 transformers (per language) after training to convergence on 10 5-SPARSE MAJORITY, 20-SPARSE MAJORITY, 5-SPARSE PARITY, and 20-SPARSE PARITY languages each. The parameters of all models are initialized uniformly. Frequencies are shown on a log scale to improve visibility. **Bottom**: Distribution of average sensitivities of 50 transformers (per language) after training to convergence on 10 5-SPARSE MAJORITY, 20-SPARSE MAJORITY, 5-SPARSE PARITY, and 20-SPARSE PARITY languages each. The parameters of all models are initialized uniformly. The $y$-axis is clipped to visualize better the higher-sensitivity outliers.

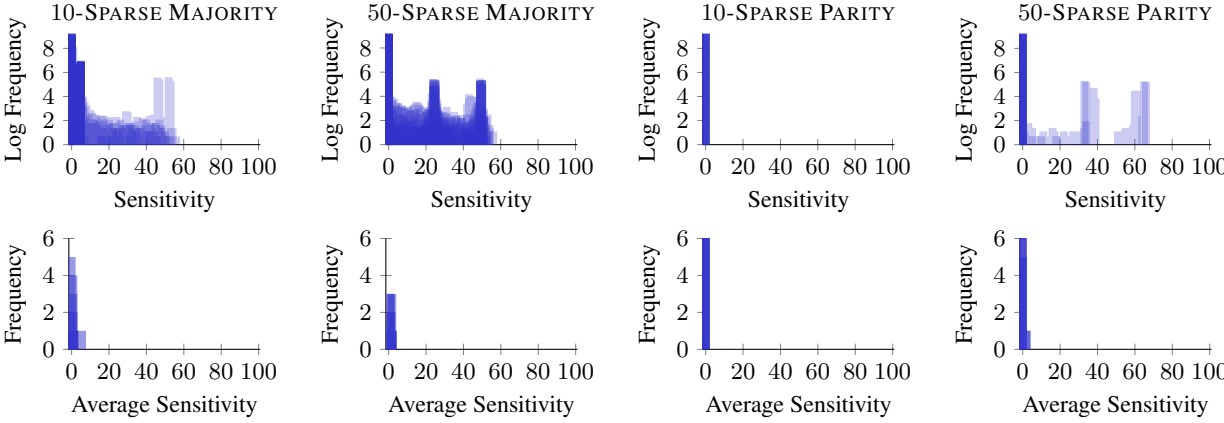

*Figure 7.* **Top**: Sensitivity spectra of 50 transformers (per language) after training to convergence on 10 10-SPARSE MAJORITY, 50-SPARSE MAJORITY, 10-SPARSE PARITY, and 50-SPARSE PARITY languages each. The parameters of all models are initialized uniformly. Frequencies are shown on a log scale to improve visibility. **Bottom**: Distribution of average sensitivities of 50 transformers (per language) after training to convergence on 10 10-SPARSE MAJORITY, 50-SPARSE MAJORITY, 10-SPARSE PARITY, and 50-SPARSE PARITY languages each. The parameters of all models are initialized uniformly. The $y$-axis is clipped to visualize better the higher-sensitivity outliers.

