# OpenReview forum: "Understanding the Parameter Space Geometry of Transformers Encoding Boolean Functions"
_ICML.cc/2026/Conference — ICML 2026 regular_

### Official Review · Reviewer_1ikU · 2026-03-08

**Soundness:** 3
**Presentation:** 3
**Significance:** 2
**Originality:** 2
**Overall Recommendation:** 3
**Confidence:** 3

**Summary:**

This paper studies the sensitivity of a random initialized transformer. Previous empirical results have shown that transformer struggles to learn high sensitivity function like PARITY. On the theory side, Hahn & Rofin (2024) showed that adding small random perturbation can cause a transformer from outputting a high average sensitivity functions to a much lower average sensitivity functions. This paper generalizes the results of Hahn & Rofin (2024) via a measure-theoretic framework and shows that a random initialized transformer, with probably almost 1, must have many input strings with low sensitivity (on sufficiently long inputs).

**Compliance With Llm Reviewing Policy:**

Affirmed.

**Final Justification:**

As explained in the "Rebuttal Acknowledgement", I'm not very strongly against with studying random initialized network now, though still have some concerns about it. I raised my score by 1 (2->3) and lowered my confidence level (4->3).

**Key Questions For Authors:**

1. Can you argue that why your results on random initialized transformer has any implications on a trained transformer.

2. The title "A Framework for Understanding Learnability in Transformers" is way too big.

**Limitations:**

This paper does have some nice discussion on their limitation. But this paper lacks of discussion on how could results on random initialized transformer have any implications on a trained transformer in the case of low-sensitivity behavior.

**Strengths And Weaknesses:**

Strengths:

1. The main theorem (Theorem 3.5) characterize the whole "sensitivity profile", i.e., for each possible sensitivity value s, theorem 3.5 gives a lower bound on the number of input string with sensitivity at most s.

2. The result in this paper looks correct, and the presentation is good.

Weakness:

1. The main weaknesses of this paper is that it only deals with random initialized transformer. This paper doesn't argue that "why random initialized transformer has low sensitivity has a strong implications for a trained transformer". I'm pretty sure one can find many properties that a random initialized transformer almost certainly satisfies, but a train transformer does not satisfy. Why low-sensitivity property is different?

2. The techniques in this paper are mostly adapted from Hahn & Rofin (2024).

---

> ### Author Rebuttal · Authors · 2026-03-31
>
> Thank you very much for your detailed review! We address your concerns below.
>
> > Can you argue that why your results on random initialized transformer has any implications on a trained transformer.
>
> Our theoretical results show that parameter settings encoding functions with positive minimum sensitivity occupy measure zero under standard initialization distributions. From a Bayesian perspective, this directly implies that the posterior is also concentrated on low-sensitivity functions: any posterior is absolutely continuous with respect to the prior and therefore inherits its measure-zero sets. While training is not exactly Bayesian inference, recent work suggests a strong connection between initialization and generalization. Buzaglo et al. (2024) [1] formalize this intuition via PAC-Bayes: the generalization capacity of a random interpolating network *after training* is tightly linked to the volume of parameter space consistent with the target function --- which our results show to be zero for high-sensitivity functions. Chiang et al. (2023) [2] provide empirical support for the relevance of this, showing that random interpolating networks generalize as well as SGD-trained ones. Dziugaite & Roy (2025) [3] strengthen the result of Buzaglo et al. further, extending it from interpolating networks to the Gibbs posterior, which softens the interpolation requirement. These results together suggest that the low-sensitivity bias we prove at initialization strongly constrains what trained transformers can generalize to: if the target function is extremely sparsely represented in parameter space, generalization towards it is very unlikely.
>
> We would be happy to include a discussion of the above in the camera-ready version if the paper is accepted.
>
> >  The techniques in this paper are mostly adapted from Hahn & Rofin (2024).
>
> While Lemma 3.2 and Corollary 3.3 are indeed adapted from Hahn & Rofin (2024), the paper employs techniques that go well beyond this. Lemma 3.4 extends the high-probability bound from a neighborhood of a single parameter setting to the full parameter space via a covering argument. Theorem 3.5 then applies the first Borel--Cantelli lemma to obtain an almost sure result holding for all sufficiently large $n$. Neither of these has a counterpart in Hahn & Rofin. Additionally, while our Lemma 3.1 is inspired by their analogous Eq. (69), the technique for bounding the key probability differs: where Hahn & Rofin use a concentration argument followed by a hyperspherical cap area bound, we express the low-variance event as a cylinder intersected with an $\ell_\infty$ ball and bound the resulting volume ratio directly, eliminating the exponential term present in their bound.
>
> > The title "A Framework for Understanding Learnability in Transformers" is way too big.
>
> We agree that the title may be somewhat misleading. Perhaps it would be more suitable to call it a framework for understanding *un*learnability. We are happy to update the title accordingly.
>
> [1] Buzaglo et al. (2024). How Uniform Random Weights Induce Non-uniform Bias: Typical Interpolating Neural Networks Generalize with Narrow Teachers. ICML 2024. https://arxiv.org/abs/2402.06323
>
> [2] Chiang et al. (2023). Loss Landscapes are All You Need: Neural Network Generalization Can Be Explained Without the Implicit Bias of Gradient Descent. ICLR 2023. https://openreview.net/forum?id=QC10RmRbZy9
>
> [3] Dziugaite & Roy (2025). The Size of Teachers as a Measure of Data Complexity: PAC-Bayes Excess Risk Bounds and Scaling Laws. AISTATS 2025. https://proceedings.mlr.press/v258/dziugaite25a.html

---

> > ### Author Rebuttal · Reviewer_1ikU · 2026-04-03
> >
> > I would like to thank the authors for their response. My concern on the significance of studying random initialized network is partially addressed. Nevertheless, I'm still not totally comfortable with studying random initialized network. E.g., the proof is basically showing that any bit-flip will not change the output of a random transformer with high probability. I feel this is qualitatively very far away from what we observed in trained LLM. Overall, I will raise my score by 1 (2->3) and lower my confidence level (4->3).

---

### Official Review · Reviewer_GKhC · 2026-03-12

**Soundness:** 3
**Presentation:** 3
**Significance:** 3
**Originality:** 3
**Overall Recommendation:** 5
**Confidence:** 2

**Summary:**

This paper give an learnability analysis of transformers. It is known that some classes of functions are hard for a transformer to learn, even though they can be expressed by it. Previous studies show that it is hard to learn functions with high average sensitivity. The paper extends the previous results to use the sensitivity profile of functions to analyze their learnability. The theoretical results imply that transformers cannot recognize functions having no zero-sensitivity strings.

**Compliance With Llm Reviewing Policy:**

Affirmed.

**Final Justification:**

This paper introduces new theoretical results aimed at bridging the gap between the expressive power and trainability of Transformers. The theory presented here effectively extends the results of previous work and, as a result, successfully provides new insights into the trainability of Transformers. While the submitted version had some issues with the readability of Section 3, the authors have adequately addressed my comments. I believe this represents a significant contribution, so I am raising the score.

**Key Questions For Authors:**

I understand that the paper presents novel theoretical results that are clearly distinct from those of Hahn & Rofin (2024). However, I'm not sure what the new results are. Is it correct to assume that results regarding the unlearnability of PARITY and FIRST were previously unknown?

**Limitations:**

Yes

**Strengths And Weaknesses:**

**Strengths**
- The paper deals with an important topic. Filling the gap between the expressibility and the learnability of transformers seems an important task.
- The proposed method extends Hahn & Rofin (2024) to show an unlearnability condition based on the sensitivity profile. The extension seems interesting, and the results it produces are valuable.


**Weaknesses**
- Section 3 is hard to follow since the connections between lemmas and corollaries are unclear.  The authors say that Lemma 3.1 is a key ingredient. However, its relationship to other results is not explained in the main body of the paper.  It would be helpful for readers if sketches of the proofs were provided.

---

> ### Author Rebuttal · Authors · 2026-03-31
>
> Thank you very much for your review! We are glad that you found our results interesting and valuable. We address your comments below.
>
> > Section 3 is hard to follow since the connections between lemmas and corollaries are unclear. The authors say that Lemma 3.1 is a key ingredient. However, its relationship to other results is not explained in the main body of the paper.
>
> We appreciate the feedback and are happy to clarify the structure of Section 3. Let us outline the connections between the results.
>
> Lemma 3.1 fixes a string $x$ and a parameter setting $\theta$, and shows that with high probability over random perturbations $\Delta$ to $\theta$, the resulting transformer has bounded blowup simultaneously on $x$ and on a set of its Hamming neighbors. Lemma 3.2 then provides the link between bounded blowup and output stability: it shows that the influence of flipping the $j$-th bit on the final output can be upper-bounded in terms of the blowup on $x$ and $x^{\oplus j}$. Corollary 3.3 combines these two results: bounded blowup implies that the output difference $|T_{\theta + \Delta}(x) - T_{\theta+\Delta}(x^{\oplus j})|$ vanishes as $n \to \infty$, with high probability. Lemma 3.4 then extends this from the neighborhood of a single parameter setting to the full parameter space via a covering argument. Finally, Theorem 3.5 aggregates over all input lengths $n$, yielding an almost sure result that holds for all sufficiently large $n$ simultaneously, rather than only for a fixed $n$.
>
> We would be happy to use the extra space available in the camera-ready version to include a more explicit high-level overview of these connections if the paper is accepted.
>
> > It would be helpful for readers if sketches of the proofs were provided.
>
> Thank you for pointing this out. We agree that proof sketches for Lemma 3.4 and Theorem 3.5 would have been good to include in the main text. We provide them below.
>
> For Lemma 3.4, the key idea is a covering argument. We cover the parameter space with essentially disjoint cubes, fine enough so that the cover's volume approximates that of $\Theta$ well. We then apply Corollary 3.3 inside each cube, and since the probability bound holds uniformly inside each, the bound extends to the full parameter space.
>
> For Theorem 3.5, the argument proceeds via the first Borel--Cantelli lemma. For each $n$, we upper-bound the probability that "too few" strings of length $n$ have low sensitivity for a random Transformer. We show that for a suitable polynomial choice of this threshold of "too few", these probabilities form a summable series. Borel--Cantelli then implies that the bound holds for all sufficiently large $n$ almost surely.
>
> We would also be happy to include these sketches in the camera-ready version.
>
> > I understand that the paper presents novel theoretical results that are clearly distinct from those of Hahn & Rofin (2024). However, I'm not sure what the new results are. Is it correct to assume that results regarding the unlearnability of PARITY and FIRST were previously unknown?
>
>
> The unlearnability of **Parity** was established by Hahn & Rofin (2024), but their results only apply to functions whose average sensitivity grows proportionally to $n$. Our framework goes beyond this in two ways. First, we take a different approach to unlearnability, shifting the focus from average sensitivity to the sensitivity profile, in particular to the number of sensitivity-zero strings. Second, this finer-grained view yields new unlearnability results for sparse parities and dictator functions --- most notably **First**, which has average sensitivity $1$ yet is provably unlearnable for long enough inputs under our framework.

---

> > ### Author Rebuttal · Reviewer_GKhC · 2026-04-04
> >
> > Thank you for the response. The authors address my concerns adequately. I have no further questions and will raise my score.

---

### Official Review · Reviewer_MR3x · 2026-03-13

**Soundness:** 3
**Presentation:** 4
**Significance:** 4
**Originality:** 3
**Overall Recommendation:** 5
**Confidence:** 3

**Summary:**

This paper studies the gap between expressivity and learnability in transformers when computing Boolean functions. It is known that transformers are expressive enough to represent highly sensitive functions such as PARITY, yet empirical evidence suggests that gradient-based training often fails to learn such functions. The paper aims to explain this discrepancy.

The main high-level takeaway is that transformers (in the specific setup considered in this paper) exhibit an inductive bias toward low-sensitivity functions, for sufficiently long inputs.

More specifically, the paper connects the sensitivity profile of a Boolean function (a fine-grained measure of how sensitive the function is to single-bit perturbations) to the class of functions that a transformer can recognize. The authors show that if all weights and biases are sampled from a standard initialization distribution (e.g., uniform or Gaussian), then with high probability the resulting function has many inputs with zero sensitivity. This suggests that during training, it is unlikely for optimization to reach parameter settings corresponding to functions that lack this property.

**Compliance With Llm Reviewing Policy:**

Affirmed.

**Key Questions For Authors:**

1. What are the main barriers to obtaining a non-asymptotic analysis?

2. How would you compare this situation to that of fully connected networks? These networks can represent functions such as PARITY, but learning them appears to require time exponential in the parity size. Would you say that such architectures are also biased toward low-sensitivity functions, or is the difficulty there primarily due to computational hardness?

3. Do you expect similar behavior for different attention mechanisms, or is the observed bias specific to the particular transformer architecture studied in this paper? Do you see a way to alleviate this inductive bias toward low-sensitivity functions?

**Limitations:**

Addressed properly.

**Strengths And Weaknesses:**

I find this paper very interesting. It studies a fundamental question about which functions transformers can express and shows that such architectures tend to represent low-sensitivity functions, characterized in a fine-grained way through the sensitivity profile. I found both the theoretical results and the experimental results to be interesting and well aligned with the main claims of the paper.

Perhaps the main limitation of the analysis is that it is asymptotic in nature, and therefore cannot fully explain some empirical observations, such as the learnability of certain classes (e.g., the class FIRST) in practice. However, these limitations are clearly acknowledged and discussed in the paper, and obtaining a non-asymptotic analysis in this setting appears to be quite challenging.

---

> ### Author Rebuttal · Authors · 2026-03-31
>
> Thank you very much for your review! We are glad to hear that you found our results interesting. We address your questions below.
>
> > What are the main barriers to obtaining a non-asymptotic analysis?
>
> All of our tools are asymptotic in nature, so we face several obstacles when trying to make them quantitative. In particular:
>
> - The probability bound in Lemma 3.1 involves a constant on the order of $(\sqrt{d})^{d-1}$, so it can only be meaningful for large $n$.
> - Lemma 3.4 requires the output difference to be small in order to imply an upper bound on the sensitivity of $x$. However, the constant hidden in $O(n^{\zeta-1})$ depends on the parameter norms within each cube of the cover, and making this uniform over all of $\Theta$ relies on compactness in a way that does not yield explicit bounds.
> - The Borel--Cantelli argument in Theorem 3.5 is inherently asymptotic: it guarantees the bound holds for all sufficiently large $n$, but gives no explicit threshold $n_0$ beyond which this is the case.
>
> Obtaining non-asymptotic results would likely require a fundamentally different set of techniques. We note that this reflects the nature of the phenomenon itself. Sensitivity-zero strings are only guaranteed to exist asymptotically, because the output difference only becomes small enough to prevent bit flips for large enough $n$. This is analogous to Hahn (2020) [1], where Lipschitzness of transformers only provably prevents output flips in the limit of large $n$.
>
>
> > How would you compare this situation to that of fully connected networks? These networks can represent functions such as PARITY, but learning them appears to require time exponential in the parity size. Would you say that such architectures are also biased toward low-sensitivity functions, or is the difficulty there primarily due to computational hardness?
>
> The difficulty of learning **Parity** and related functions in fully connected networks is a subtle question. Statistical query complexity bounds show that SGD requires exponentially many samples to learn sparse parities, but these bounds do not directly address the full **Parity** function. Recent work by Abbe et al. (2025) [2] shows that for MLPs, learnability of full **Parity** depends critically on initialization: discrete Rademacher initialization enables efficient learning, while Gaussian initialization prevents it. This suggests that the initialization-dependent bias we identify for transformers may be more broadly relevant, though a formal analysis for fully connected networks is beyond the scope of this paper.
>
> > Do you expect similar behavior for different attention mechanisms, or is the observed bias specific to the particular transformer architecture studied in this paper?
>
> We believe our arguments are fairly generic to transformer architectures. The proof relies on the properties of layer normalization rather than the specific attention mechanism. We expect popular variations such as RoPE, different placements of layer norm, or RMSNorm to leave our conclusions intact.
>
> > Do you see a way to alleviate this inductive bias toward low-sensitivity functions?
>
> Due to the generality of our results, we expect that alleviating this bias would require going substantially beyond standard self-attention. The most promising direction is perhaps hybridizing transformers with state space models (SSMs) with negative eigenvalues.
>
> [1] Hahn (2020). Theoretical Limitations of Self-Attention in Neural Sequence Models. TACL 2020. https://aclanthology.org/2020.tacl-1.11/
>
> [2] Abbe et al. (2025). Learning High-Degree Parities: The Crucial Role of the Initialization. ICLR 2025. https://openreview.net/forum?id=OuNIWgGGif

---

> > ### Author Rebuttal · Reviewer_MR3x · 2026-04-04
> >
> > Thanks for yout response.

---

### Official Review · Reviewer_65es · 2026-03-17

**Soundness:** 4
**Presentation:** 3
**Significance:** 2
**Originality:** 3
**Overall Recommendation:** 5
**Confidence:** 4

**Summary:**

The paper analyze typical transformer that are randomly initialized either with a uniform distribution over a compact set or with Gaussian distribution. By doing so, they proved that almost every such transformers have at least a polynomial number of string inputs on which they are stable, when $n$ is large enough. As an immediate corollary, the paper derives lower bounds on the improbability of random transformer to solve tasks where a lot of strings are highly sensitive (e.g. parity, k-juntas, etc.). Some experiments are performed to verify sensitivity metrics of initialized, as well as trained transformers

**Compliance With Llm Reviewing Policy:**

Affirmed.

**Final Justification:**

Concerns are addressed in the rebuttal

**Key Questions For Authors:**

- What are the main difficulties in extending these results to variable length transformer, with a proper normalized definition of 'sensitivity'?
- Can the proven results be obtained with a simpler spectral analysis, using some kind of random matrix theory at initialization?

**Limitations:**

See Weaknesses

**Strengths And Weaknesses:**

Strengths:
- The paper main contribution is theoretical, and the proofs are theoretically sound, as far as the reviewer can check (appendix' proofs are not checked carefully but the reviewer believes such results are plausible; derivations in the main text were checked)
- The transformer settings used in the paper is interesting and might benefit the theoretical community, since choosing a suitable transformer model to obtain mathematical guarantees are not trivial.
- The experiments performed well-support the main claims of the paper, that transformer tends to bias towards functions that have lower sensitivity, in many metrics.

Weaknesses:
- The setting of the paper is rather restrictive, in that the authors could only obtained high probability results for random transformers. If one just randomly initialized a transformer without training, chances are they will not be able to be accurate on very complex, spiky functions. Although it is still interesting to quantify the previous 'folklore', having a guarantee of polynomially many (for constant $k$) stable strings (over an exponential number of total possible strings), is not a very strong bound. (The authors are welcome to contest on this point if there is an argument to show that the lower bound is 'tight' to some extent.)
- The setting of the paper for fix $n$ is also restrictive. While it would generally be harder to have a variable-$n$ training datasets, as the paper does not deal with training, there isn't a natural justification to fix the length of the input string. If this were a common assumption in the literature of this types of results, the authors should cite such precedence.
- Minor: The authors, after deriving their theoretical results, claim that transformer is biased towards functions with low sensitivity. However, this is not immediately obvious to me, since there could be much more low-sensitive functions than high-sensitive ones (in which case, even if the transformer is unbiased and uniformly sample from all such Boolean functions, they could have higher probability of getting a low-sensitive one).
- Minor: The experiments also run rather adversarial to the main message of the paper, since they show that with training, transformer could learn to to be less bias towards low-sensitivity function. While the paper does not make any guarantees for trained transformer, the problems raised as 'limitation' or 'learnability' of transformer seems to go away with training, giving the impression that the problem isn't much of a problem. The authors can perhaps still identify and discuss some limitations with these biases, even with trained model; or derive an intervention (e.g. regularizer, different loss function) that debias the model away from low sensitive functions).

---

> ### Author Rebuttal · Authors · 2026-03-31
>
> Thank you very much for your detailed review!
> We address your concerns in the following.
>
> > The results only apply to randomly initialized transformers, not trained ones.
>
> Reviewer 1ikU raises the same point, and we address it in detail in our response to them.
>
> > The bound of polynomially many stable strings is not very strong.
>
> While we agree that the bound may not be tight, we note that it is sufficient to prove interesting properties for a broad class of languages, including Parity, sparse parities, and dictator functions.
>
>
> > What are the main difficulties in extending results to variable-length transformers?
>
> We believe there may be a misunderstanding here. In our final results (Theorem 3.5, Corollaries 3.6 and 3.7), we do not fix $n$. We consider Boolean functions $f\colon \lbrace0,1\rbrace^* \to \lbrace0,1\rbrace$ and transformers as functions $\lbrace0,1\rbrace^* \to \lbrace0,1\rbrace$, and prove statements about their sensitivity profiles (Definition 2.4) for all sufficiently large input lengths $n$ --- namely, that transformers must have sensitivity-zero strings for all such $n$. The definition of Boolean functions as $f\colon \lbrace0,1\rbrace^n \to \lbrace0,1\rbrace$ for fixed $n$ appears only as an intermediate step, which is standard in the Boolean function complexity literature (O’Donnell, 2014 [1]). We will clarify this more prominently in the paper.
>
> > Is the claimed bias simply a reflection of there being more low-sensitivity Boolean functions than high-sensitivity ones?
>
> This is a very good point, since it is indeed true that there are more low-sensitivity Boolean functions than high-sensitivity ones. However, the Transformer's preference extends beyond this, as illustrated by the following computation.
>
> Consider a uniformly random Boolean function $f_n\colon \lbrace0,1\rbrace^n \to \lbrace0,1\rbrace$, i.e., $f_n(x)$ is sampled with equal probability from $\lbrace0,1\rbrace$, independently for each $x$. Then any given input $x$ has sensitivity $0$ with probability $2^{-n}$. It can be shown that the number of sensitivity-zero strings converges in distribution to a Poisson(1) random variable, so the probability that at least one such string exists is approximately $1 - 1/e \approx 0.63$ for large $n$. Meanwhile, a random transformer computes a function with sensitivity-zero strings with probability 1 for large $n$. This asymmetry means the inductive bias toward low-sensitivity is a genuine structural property of the architecture, not merely a reflection of the prior over Boolean functions. We are happy to add a discussion of this to the paper.
>
>
> > The experiments seem to show the bias is reduced with training.
>
> Our experiments actually show that the sensitivity bias persists after training --- it may appear to diminish in Fig. 2, but this is partly an artifact of the log scale used in the sensitivity spectra. The vast majority of strings still have sensitivity zero or near zero after training. This is corroborated by Bhattamishra et al. (2023) [2], whose experiments demonstrate that Transformers prioritize learning functions of low sensitivity.
>
> > Could the authors identify interventions to alleviate the bias in trained models?
>
> For a note on potential interventions, we refer the reviewer to our response to Reviewer MR3x.
>
>
> > Can the results be obtained via spectral analysis or random matrix theory?
>
> This is an interesting direction, but several technical obstacles make a straightforward spectral/random matrix theory approach difficult. Any such analysis needs to take into account two key issues: First, the attention mechanism is nonlinear and not straightforwardly handled by random matrix theory. Second, the effect of layer norm in the $\varepsilon \to 0$ regime breaks the smoothness of the architecture, so one cannot simply combine random matrix theory with smoothness of the nonlinear parts. A heuristic argument based on linearization might be possible via frameworks such as Tensor Programs, but we believe this would not formally accommodate the $\varepsilon \to 0$ regime, which breaks pointwise Lipschitzness of the transformer model. In a weaker model where layer norm is Lipschitz ($\varepsilon$-bounded away from $0$), a spectral approach may be feasible, using high-probability bounds on matrix norms combined with smoothness of the nonlinear components --- but at the cost of a weaker model of transformer expressivity. We leave a formal investigation of this for future work.
>
> [1] O'Donnell (2014). Analysis of Boolean Functions. Cambridge University Press.
>
> [2] Bhattamishra et al. (2023). Simplicity Bias in Transformers and their Ability to Learn Sparse Boolean Functions. ACL 2023. https://aclanthology.org/2023.acl-long.317

---

> > ### Author Rebuttal · Reviewer_65es · 2026-04-04
> >
> > 1. Random initialization. The points that the authors raised in the rebuttals to 1ikU is interesting but I believe that years of deep learning optimization trajectory studies have shown that even a measure 0 set of exception is not good enough (e.g. 0-loss manifold studies are over measure 0 sets). However, I agree that this is the current limitation of the field and would take some serious breakthroughs to overcome so this point is not unique to this paper.
> > 2. The other points are sufficiently addressed, for a theory paper. I have raised my score accordingly

---

### Decision · Program_Chairs · 2026-04-30

**Decision:**

Accept (regular)

**Comment:**

This submission studies the functions represented by transformers with randomly initialized weights and establishes a strong bias toward low-sensitivity functions. The reviewers appreciated the solid technical contribution, but they also raised concerns regarding the restrictive scope of the theoretical setting, in particular that the theory applies only to transformers with random weights. The meta-reviewer therefore strongly suggests that the authors revise the main text and highlight this limitation in the title.